# Creative practice as a tool to build resilience to natural hazards in the Global South

Anne F. Van Loon[1], Imogen Lester-Moseley[2], Melanie Rohse[3], Phil Jones[2], and Rosie Day[2]

[1]Institute for Environmental Studies (IVM), Vrije Universiteit Amsterdam, De Boelelaan 1087, 1081 HV Amsterdam, The Netherlands
[2]School of Geography, Earth and Environmental Sciences, University of Birmingham, Birmingham, UK
[3]Global Sustainability Institute, Anglia Ruskin University, Cambridge, UK

**Correspondence:** Anne Van Loon (anne.van.loon@vu.nl)

**Abstract.** Global South communities are increasingly exposed and vulnerable to natural hazards such as floods and droughts. Preparing for future hazards requires developing an idea of an uncertain future, thinking out of the box for possible solutions, enhancing communication between diverse groups, and instigating organisational and behavioural change. In this study, we explore whether art and creativity could support this process, by presenting a literature mapping and a case study. Our search for journal articles, focusing on Global South communities and topics like environmental issues, hazards, and health, yielded 267 papers published between 2000 and 2018. These used a diversity of art forms, including photography & other forms of visual art, music & song, and drama & storytelling. We found that papers on the topic of climate change generally had lower co-creation (62% medium to high) than those on health (90% medium to high). A subset of seven papers focusing on drought and flooding fell into two categories: those aiming to raise the general public's awareness of these hazards and those aiming to instigate adaptation action by the participants. In our case study, we explored the middle ground between these categories. In a pilot project in South Africa, we designed storytelling workshops, in which community members explored scientific data on future droughts, exchanged ideas between groups, and developed narratives about impacts of and preparedness for future drought. These narratives were filmed and edited and shared both with the community and with governance actors. We found that this approach allowed participants to imagine future droughts, it opened up conversations about potential adaptation measures, encouraged intergenerational exchange, and increased awareness of local issues by policy makers. Both in the wider literature and in our case study, the long-term effects of creative interventions are rarely evaluated. Feedback from participants, however, indicates a number of short-term benefits, which shows the potential of combining creative practice approaches and more conventional approaches into a more holistic preparation for future natural hazards.

## 1 Introduction

Global South communities are vulnerable to the impacts of hazards like floods and droughts, and are expected to be even more at risk in the future (Winsemius et al., 2015b), as increased climate variability is likely to lead to more floods and droughts (IPCC, 2012) and water demands, exposure and vulnerability are growing (Wanders and Wada, 2015; Winsemius et al., 2015a). Better resilience and preparedness to floods and droughts are urgently needed. Preparing for future extremes

requires including diverse knowledges, elevating under-represented voices, thinking out of the box for possible solutions, enhancing communication between diverse groups, and instigating organisational and behavioural change. In this paper, we investigate how creative and art-based methods can support this transition to more resilience to natural hazards, and more specifically droughts and floods, in Global South communities. To achieve that aim we systematically map the literature on how art and creativity are used in relation to floods and droughts and other natural hazards, and in related fields (climate change, health), and further discuss a specific case study of our own as an exemplar of using creative practice to increase resilience to drought.

We will use the word 'creative practice' to mean all artistic and creative processes (Niedderer and Roworth-Stokes, 2007). Where creativity can be seen as the production of a novel and appropriate response to a particular concern (Sternberg, 1999), creative practice is commonly associated with arts-led techniques. The term stretches to cover writing, theatre, dancing and a host of other imaginative activities (Light, 2018), not all of which need to result in a conventional product of aesthetic merit (Field, 1950). Games could be classed as creative practice because they might include an element of creative exploration of possible options. However, in this study, we exclude games, but point to some excellent overview papers on how 'serious games' can be used for disaster risk management (Solinska-Nowak et al., 2018), climate change adaptation (Flood et al., 2018), environmental management (Madani et al., 2017; Aubert et al., 2018), and flood management (e.g. Ramos et al., 2013; Crochemore et al., 2016; Arnal et al., 2016).

By 'Global South communities' we mean poor, vulnerable, less-powerful communities living in the geographic South as well as North (Barreto, 2014, p.404). We use the term Global South, recognising that this has developed from the merely geographical to more of a political and economic characterisation. It is commonly used to refer to, but is not completely overlapping with, issues of inequality, power imbalance, and deep relative poverty. It therefore encompasses a variety of vulnerable and socio-economically disadvantaged groups, including much of the rural populations in the geographic South, those in informal settlements, and groups who are marginalised because of race, gender, age. We note that the geographic South also contains privileged and wealthier communities who might better be classed as global North (Mahler, 2018). In our mobilisation of the definition, we have also included indigenous communities, refugees, and children and young people located in the geographic North, although have not stretched as far as to include more socio-economically deprived communities of the geographic North.

We focus on Global South communities because they often do not have access to large-scale structural (i.e. engineering-based) mitigation options such as dikes protecting against flooding or reservoirs to overcome dry periods, either because these are too expensive or considered poor 'value for money' or because they are not feasible in the region these communities inhabit (e.g. Johnson and Priest, 2008; Ikeda et al., 2016). Also, these communities might have knowledge of suitable traditional structural or non-structural measures, such as qanats (water capturing systems), soil management, food storing systems, social support mechanisms (Barontini et al., 2017; Berkes et al., 2000; Altieri and Nicholls, 2013). In both cases, there is a need to surface their hidden voices and to explore which measures work best in the local context. According to Gibson and Gordon (2018), cultural resourcefulness and coping capacities of rural populations are rarely acknowledged within state-expert modelling of resilience.

There is an important body of literature that critiques the term 'resilience', alerting us to the need to use it cautiously (e.g. Davoudi et al., 2012; Moser et al., 2019). For example, MacKinnon and Derickson (2012) argues that resilience could be used by people in power to denote 'self-reliance', thereby putting the onus of risk management on individuals or communities that do not have the means and power to effectively achieve this. We are acutely aware of the sensitivity of our research topic, with the risk of marginalised communities being denied access to structural measures and potentially being offered creative practice as an alternative. Therefore, we use the term 'building resilience' to denote addressing and mitigating the complex interactions of social and economic vulnerability of communities and supporting their way of preparing for, coping with and recovering after disasters. We focus on the added value of creative practices alongside more established processes of resilience building and we explicitly include decision makers in this investigation, to also study the added value of creative practice for those with more agency.

In the following sections, we first map the existing literature on this topic and identify research gaps (Sect. 2). Then, we discuss methods and results of a case study (Sect. 3), indicating potential for the research gaps to be filled. Finally, based on the literature mapping and the case study, we share reflections and perspectives for future research (Sect. 4). We see potential for creative methods as a currently under-explored way to give voice to marginalised communities and to empower them to take action or seek support to increase preparedness to extreme events. We envisage creative methods to be part of a port- folio of methods to build community resilience to hazards and call for more research on the effectiveness of these tools in communicating about flood and drought risk or as a way for communities to imagine future risk or potential preventive actions.

## 2 Literature

### 2.1 Literature framework

Research on art and creativity to produce change shows that these are used by a range of people in different ways with a variety of goals, participants, and audiences. There are different ways to categorise such art-based practice. Miles (2010), who studied art exhibitions on the topic of climate change distinguish two types of aims: raising awareness and intervention. Guba and Lincoln (1989) adds the dimension of the audience or receiver: is the creative process or art product enhancing the insights of the participants or of others? Rathwell and Armitage (2016) noted the same categories, but added the aspect of the experience of the artistic process, noting 'art as a site of knowledge coproduction' (Rathwell and Armitage, 2016, p.1).

From these categorisations, three dimensions emerge that characterise creative practice to produce change in / with / for communities: the goal of the creative practice, the doer, and the audience. The goal of this type of creative activity can be to raise awareness, instigate action, or both. The doer of the creative practice then refers to whether the creative practice is carried out by community members, or by an academic or artist, or whether it is co-developed by community and academics / artists. The intended audience of the artistic product or those who benefit from the creative practice can be the participants themselves, or other community members, decision makers, the general public, or researchers. The existing literature on creative practice used by or with communities shows these three dimensions in various combinations.

First, there are examples of where the creative practice is community-led, with other community members as the audience. Research on Traditional Ecological Knowledge (TEK[1]) shows how TEK is often passed on within communities using traditional stories, songs, dance, etc. Researchers have described and documented these methods doing participant observation. For example, Rigby et al. (2011) and Zurba and Berkes (2013) show how art is used by aboriginal communities to (re-)connect to the land in periods of environmental stress, McEwen et al. (2012) describe the practice of archiving diverse flood information including narratives and songs with the aim to provide a rich recourse to communities living with flooding, and there are various examples of how traditional songs and stories are used to pass on knowledge between generations (Moncada, 2018; Simpson, 1999). In these cases, the goal of the creative practice can be both raising awareness or instigating action. Also interesting to note is that during this kind of research no new material is created and all creative practice happened before the researchers step in.

Second, art therapy (Rubin, 2016; Slayton et al., 2010) is an example of creative practice used with the goal to instigate action (or behavioural change) and done by the participants. The use of art and creativity in a therapeutic way has been studied extensively (Snyder, 1997; Edwards, 2014), for instance as a post-disaster recovery and healing therapy (e.g. Huss et al., 2016; Zerrudo, 2016; Whittle et al., 2012). With regard to droughts and floods more specifically, there is some evidence from Australia that art and music festivals provide an escape from the hardship of prolonged drought, bring the community together, and enhance emotional well-being (Gibson and Connell, 2015). Here, the creative activities are used to forget the economic impacts of environmental issues or natural hazards and lessen their social impacts (e.g. feelings of isolation, loss of community, depression, suicide; Gibson and Connell, 2015). In other examples, psychological impacts are lessened by using artistic processes to more deeply explore feelings and experiences (Whittle et al., 2012). In this form of art therapy, traditional creative methods can be used or new ones chosen by the therapist, and both the doer and the audience are the individual participant or the wider community. A review by Rubin (2016) found that there is 'quantifiable data to support the claim that art therapy is effective in treating a variety of symptoms, age groups, and disorders' (Rubin, 2016, p.108).

Third, art and creativity can be used as an active process (where new material is created during the research) with the goal of instigating action or behavioural change in the participants in a broader sense. This is, for example, studied in education (Bequette, 2007; Silo and Khudu-Petersen, 2016; Cramer et al., 2017) and health (Schmid, 2006). In these cases, researchers often have a more active role in guiding the process, sometimes in collaboration with artist(s). Again, traditional art forms can be used or built on or new art forms can be proposed by the researcher. There is an emerging literature on using art in this way to build social-ecological resilience (Rathwell and Armitage, 2016) or to deal with floods and droughts (Mason, 2015). Like in art therapy, the focus is on participants as the audience, but there is more involvement from the researcher in this category.

In a fourth category, the aim is scientific awareness (or creating new knowledge) and the audience of the creative process are the researchers themselves and the scientific community. If creative practice is used as research tool, the aim often is to reach deeper layers of people's lived experience of environmental issues or natural hazards (Skains, 2018). In this case, the process is used by the researcher(s) as a qualitative data collection method to increase their understanding and knowledge on the topic

---

[1]TEK refers to 'all types of knowledge about the environment derived from the experience and traditions of a particular group of people'. (Usher, 2000, p.185)

(e.g. Kloetzel, 2017; Miller and Brockie, 2015). Using art and creativity in this way has been argued to give vulnerable people a voice and to allow the message and emotions to travel beyond those who experienced the event, but there seems to be no empirical research confirming this (Miller and Brockie, 2015).

Finally, there are examples of art and creative processes used with the goal of raising awareness of the general public or instigating behavioural change of large groups of people. In those cases, the doer can be an artist or members of the general public themselves. Researchers have investigated how various organisations (NGOs) and artists have used creative practice in public-facing endeavours (e.g. Curtis et al., 2012). These creative 'interventions' can have a range of aims, including communicating to an audience about environmental issues, raising awareness, reshaping public perceptions, enhancing engagement, and promoting action (Rice et al., 2019). On the topic of climate change, for example, art is often used with a focus on the general public, for communication and awareness raising (Nurmis, 2016) and instigating behavioural change (Burke et al., 2018). In some of these the audience is quite passive, but there are also examples of how the public is engaged in participatory art (Candy et al., 2006). The reasons for using art as an engagement tool include that it can help people understand complex information (Curtis et al., 2012), can support the development of new mental models, changing paradigms and beliefs (Lozano, 2011), and is a powerful way to make people care about a topic because it can invoke strong emotions (Matravers, 2001; Silvia and Brown, 2007; Barbour and Hitchmough, 2014). Interestingly, existing studies on the effectiveness of art-based climate change communication offer only limited and inconsistent evidence of their impact. Some researchers also mention a potential use of artistic products in decision making (e.g. Symons, 2016), but to our knowledge no published examples of this exist.

There is of course overlap between these types and studies often do not fall only in one category. For example, when participants are the audience (when archiving TEK or in art therapy, when the aim is instigating action), the results can travel to others in the community and to policy makers and when creative practice is used as a research tool, also policy makers or the general public could read the academic papers. In our literature review, we will focus on the primary audience and aim, but discuss mixed cases and secondary audiences and aims as well. In most of these examples new material is created during the research process, except for the first example of documenting TEK, in which the material was already created before the research and creative practice was done without the involvement of the researcher (making co-creation impossible).

In this paper, we will focus on all these combinations of the three dimensions of using creative practice (goal, doer and audience), excluding its therapeutic use. There is already much research on art therapy (e.g. Rubin, 2016; Slayton et al., 2010), whereas here we are mostly interested in how art-based information can be used to make voices heard, enhance communication between diverse groups, think out of the box for possible solutions, and instigate organisational and behavioural change.

## 2.2 Literature mapping

We mapped the anglophone academic literature to find papers reporting on uses of creative practice to raise awareness or build resilience to drought and flood risk in the Global South. We also looked at other hazards (earthquakes, volcanic eruptions, etc.) and environmental issues, including climate change, more generally. As there is a more established praxis of using creative practice in instigating behavioural change in health-related issues (notably HIV), social and economic inequality, and violence

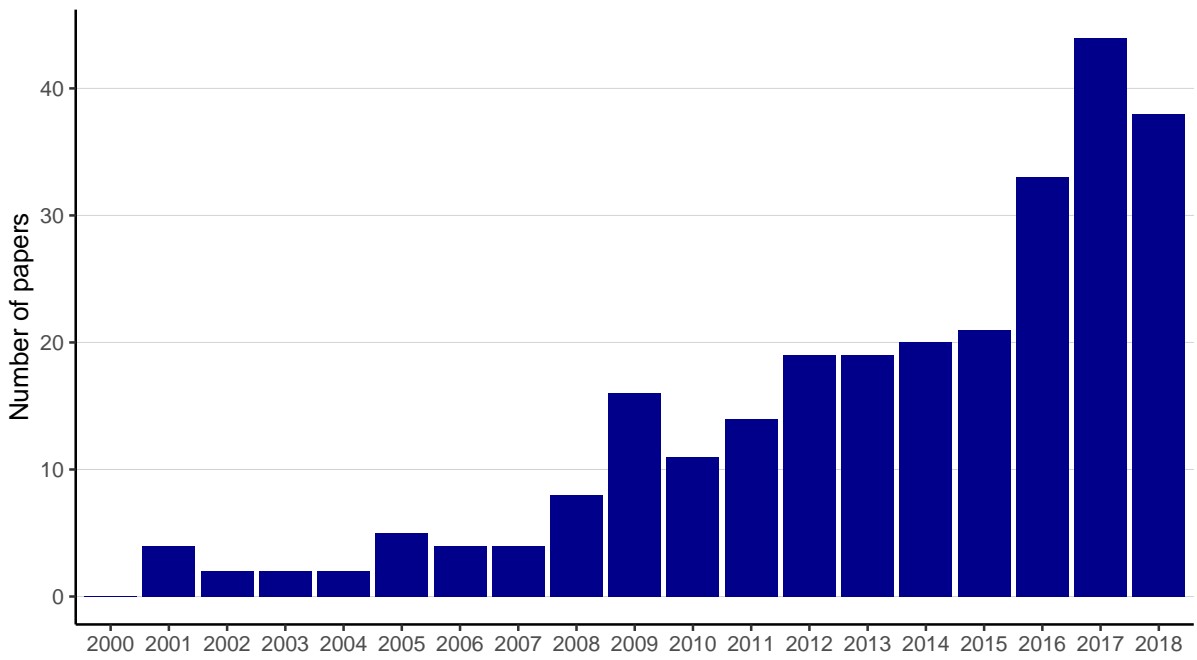

**Figure 1.** Peer-reviewed journal articles (in no. per year) found via a literature mapping exercise, focusing on the use of art-based and creative practice in the research fields of hazards and disasters, climate change, other environmental issues, health, social and economic inequality, and violence & conflict in Global South context (for details see Appendix A).

.

and conflict, we have also reviewed the literature on those topics. More details on the literature mapping exercise can be found in Appendix A.

Our iterative search process with manual screening resulted in a selection of 267 journal articles. These show a clear increase in number per year over time, especially after 2008 (Fig. 1), which is consistent with reviews focusing on art and climate change (Galafassi et al., 2018) and art and vulnerable populations (Coemans and Hannes, 2017). Most of the papers focus on topics related to health (21% of total no. of papers) and climate change (18%); just over 15% discuss the use of creative practice in topics of environmental management and resource access. Papers on using creative practice in relation to natural hazards and disasters (volcanic eruptions, earthquakes, tsunamis, drought, and flooding) only make up 1-3% of the total sample each.

In this study, we searched for papers focusing on Global South communities, encompassing vulnerable, less-powerful groups in society. The resulting selection was spread across the globe (Fig. 2). In the results of our search, most of this type of research is done in Africa (29% of total no. of papers), followed by Oceania (25%) and Asia (16%). There is also a lot of work with indigenous communities and vulnerable groups (refugees / asylum seekers, children / young people) in North America (16%), but only a small amount in Europe (3%). Recent more generic review articles also found that research on arts and health is often done in Africa (Teti et al., 2018) and water-related creative practice research is concentrated in water-scarce regions in Africa

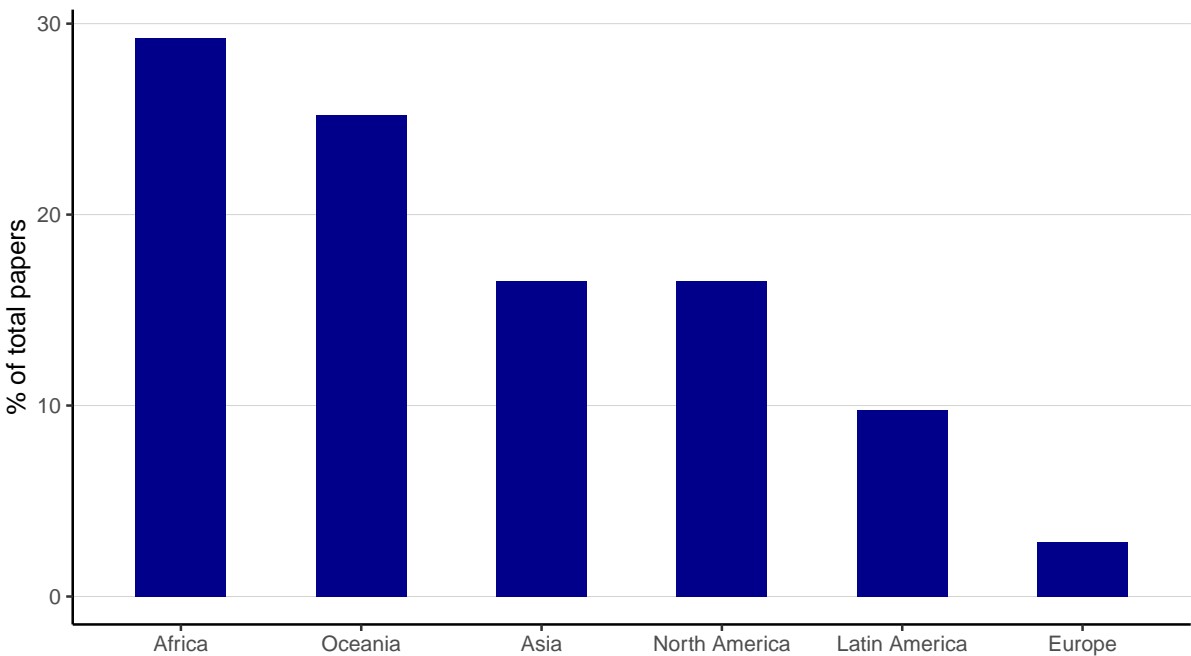

**Figure 2.** Continents (in % of total papers) where the research was carried out that we found via a literature mapping exercise (for details see Appendix A).

and Australia (Fantini, 2017). Other reviews on the use of different art-based methods in environmental and health research found that most art-based research is carried out in the United States, Canada, the United Kingdom and France (Nurmis, 2016; Galafassi et al., 2018; Coemans and Hannes, 2017).

The creative methods and art forms used are very diverse (Fig. 3). Photography is the most-used method (mentioned 63 times), followed by music & song (59 times), other forms of visual art (48 times), drama (46 times), storytelling (43 times), and video & film (37 times). Art forms such as dance and poetry are used less (17 and 12 times, respectively). One reason for the large amount of papers on photography is that it is often used as a participatory research methodology in which participants can highlight issues of importance to them. Photovoice (or Photo-Elicitation Methods or Camera-User-Study) is a participatory

method that asks community members to photographically document their environment or situation and share stories about these photographs. In our literature review, we found that Photovoice has often been used to explore communities' view on HIV / AIDS (e.g. Jacobs and Harley, 2008; Mitchell et al., 2005; Umurungi et al., 2008; Wood, 2012; Fournier et al., 2014; Adegoke and Steyn, 2017), environmental issues (e.g. Belcher and Roberts, 2012; Bennett and Dearden, 2013), conservation (e.g. Beh et al., 2013), water use and governance (e.g. Fantini, 2017; Bisung et al., 2015), and hazards and disasters (e.g. Yoshihama and Yunomae, 2018; Schumann et al., 2018). This means that photography is mostly used as a research method

asking participants to develop new material. Papers on music and song, on the other hand, mostly study existing traditional songs and music on a variety of topics (e.g. Stone, 2003; Saroli, 2005; Wu, 2016; Grant, 2018; Dirksen, 2019). They are rarely

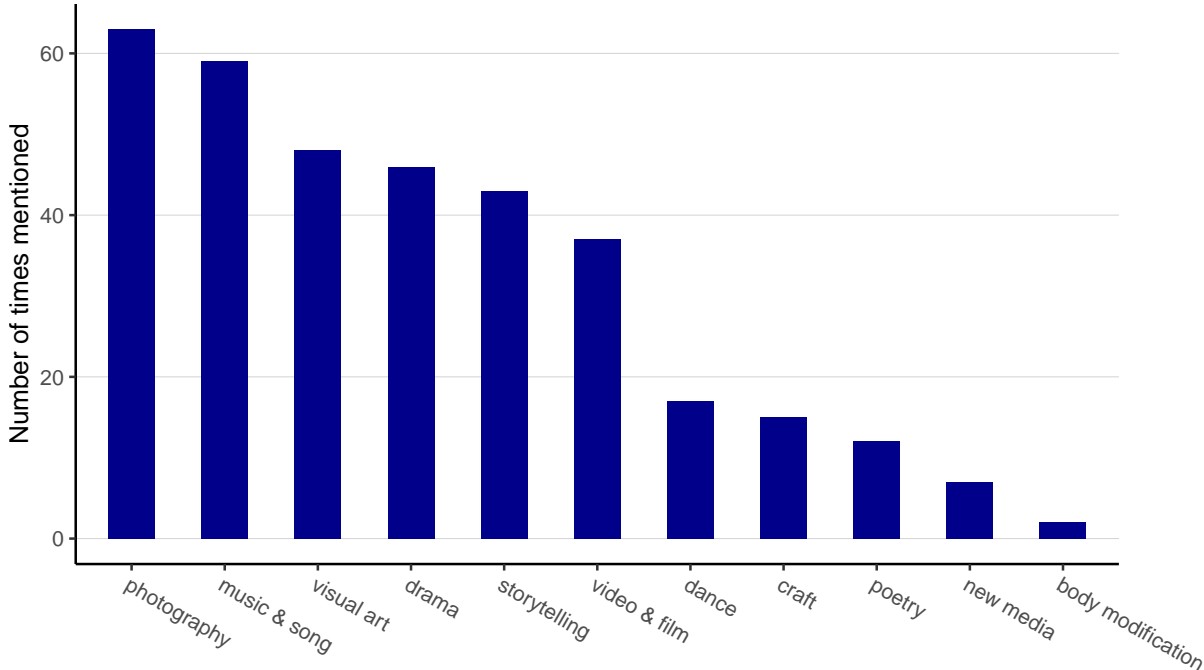

**Figure 3.** Art forms / creative methods used or researched (in no., multiple art forms per paper possible) in the journal articles that we found via a literature mapping exercise (for details see Appendix A).

used to co-create new material; some examples where new music is created are: Steiner (2015), Anderson et al. (2018) and Plush and Cox (2019).

We also looked at the degree of co-creation between the researchers and communities, because we were interested to see how much the researchers were involved in the creative practice, including initiating, supporting, guiding or even leading the creative practice, and how much of the creative practice was pre-existing in the community prior to the research or was completely carried out by the community without researcher involvement. The degree of co-creation is very variable between papers. In some cases, the methodology was given to a community by researchers (for example in participatory photography), but the resulting product was made without the researcher's involvement (e.g. Belcher and Roberts, 2012). In other cases, the participants communicated to the researchers what the greatest environmental threats to their community are, and then artists developed this into an artistic product (e.g. Steiner, 2015). There are also many examples of researchers observing and documenting creative practice traditionally used in Global South communities to pass down traditional knowledge on natural hazards or environmental issues, for example using storytelling (e.g. Swanson, 2008; Cashman and Cronin, 2008; Johnson and Beamer, 2013; Troll et al., 2015; Fepuleai et al., 2016). These have a low degree of co-creation, because they work with existing material, often in a form of participant observation. Others build on traditional creative practice to develop new stories (e.g. Fuertes, 2012; Somerville, 2014; Ayala et al., 2016; Fernández-Llamazares and Cabeza, 2017). These often have a high degree of co-creation with the researcher guiding the process.

The aim of the art-based research also strongly affects the degree of co-creation. In many papers, the goal of developing an artistic product is communication, for example to raise awareness of an environmental issue and its impact on vulnerable communities. There is a wealth of projects aiming at raising awareness of the general public on climate change and its impacts (Nurmis, 2016; Galafassi et al., 2018). Papers on the topic of climate change generally have a slightly lower degree of co-creation (62% medium or high, compared to 67% medium or high for all papers included in the literature review; see Appendix A). In other research, the goal of the creative practice is instigating some kind of action, for example to bring about behavioural change in relation to health (for example HIV / AIDS). Papers on the topic of health generally have a higher degree of co-creation (90% medium or high, compared to 67% medium or high for all papers included in the literature review).

Only rarely is the effect of the creative practice evaluated. Fantini (2017) highlights that creative participatory methods such as Photovoice are claimed to be effective in communicating community concerns, but that empirical evidence for these claims is missing. There are some examples where results are reported. In Contreras et al. (2018), theatre-based workshops were a significant success, encouraging almost half of the participants to seek out government-provided health services after a disaster. However, a comparison with a control case, in which other, non-creative methods were used or in which no activity was done at all, is hardly ever done. This important aspect will be further explored in Section 4.

In our search, we found only two papers focusing on drought and/or water scarcity (Table 1: D1 and D2). Barontini et al. (2017) used arts as a communication tool. They documented traditional irrigation techniques to cope with water scarcity in the Mediterranean and, together with other researchers and students, developed a travelling exhibition for students and the general public. In this example, the exhibition was used to raise awareness and understanding of historical water conservation techniques. In contrast, Rigby et al. (2011) used arts as a tool to change behaviour and coping capacity. They investigated the use of creative and artistic practice in response to drought and discuss how encouraging Aboriginal arts in Australia can increase resilience to drought as it enhances the connection with the land. They mention a whole suite of art forms used traditionally to highlight the Aborigines' connection to land, e.g. painting, printing, photography, film, theatre, music and dance. This research did not develop new artistic products or ask participants to engage in new forms of creative practice. Instead, they studied how traditional art can help people cope with drought by using existing material previously developed by the community (so without co-creation).

Five papers focusing on flooding were identified (Table 1: F1-F5). They ranged from studies on using community workshops to merge scientific and local knowledge of flooding (Ikeda et al., 2016) to a researcher / poet developing poems from interview transcripts of older people's lived experience of flooding (Miller and Brockie, 2015). Three studies used participatory creative methods: participatory theatre to help young people cope with mental health issues related to flooding post-disaster (Contreras et al., 2018), participatory photography to explore questions of flood management (Stephan, 2018), and participatory drawing to understand flood impacts on young children (Mort et al., 2018).

In Figure 4, we have classified these seven papers on drought and flooding (Table 1) following the three dimensions introduced in Sect. 1: i) the goal of the creative practice (raising awareness, instigating action, or both), ii) the doer of the creative practice / the creator of the end-product (completely led by the community, an academic or artist, or co-created between community and academics / artists), and iii) the audience of the creative practice / the end-product (the participants themselves,

**Table 1.** Papers on drought (D1, D2) and flooding (F1-F5) found in the literature mapping exercise focusing on creative practice and Global South communities, and the case study (CS) described in Sect. 3

| No. | Title | Author(s) & Year | Journal | Region |
|---|---|---|---|---|
| D1 | Bridging Mediterranean cultures in the International Year of Soils 2015: a documentary exhibition on irrigation techniques in water scarcity conditions | Barontini et al. (2017) | Hydrology Research | Europe, Africa |
| D2 | If the land's sick, we're sick:* the impact of prolonged drought on the social and emotional well-being of Aboriginal communities in rural New South Wales. | Rigby et al. (2011) | The Australian Journal of Rural Health | Oceania |
| F1 | Knowledge Sharing for Disaster Risk Reduction: Insights from a Glacier Lake Workshop in the Ladakh Region, Indian Himalayas | Ikeda et al. (2016) | Mountain Research and Development | Asia |
| F2 | The disaster flood experience: Older people's poetic voices of resilience | Miller and Brockie (2015) | Journal of Aging Studies | Oceania |
| F3 | Community strengthening and mental health system linking after flooding in two informal human settlements in Peru: a model for small-scale disaster response | Contreras et al. (2018) | Global Mental Health | South America |
| F4 | Social practices of flood (risk) management – a visual geographic approach to the analysis of social practices in an empirical case in Chiapas, Mexico | Stephan (2018) | Erdkunde | Central America |
| F5 | Displacement: Critical insights from flood-affected children | Mort et al. (2018) | Health & Place | Europe |
| CS | Hydrological modelling as a tool for interdisciplinary workshops on future drought | Rangecroft et al. (2018) | Progress in Physical Geography | Africa |
| | Using a narrative method to imagine preparedness to future droughts in South Africa | Rohse et al. (in prep.) | Geo: Geography and Environment | Africa |

other community members, decision makers, the general public, or researchers). Figure 4 shows that the papers generally fall into two categories related to the goal and audience dimensions. Firstly, those discussing creative practice aimed at com-
municating the impacts of drought or flooding to the general public (D1, F2) or to researchers (F4, F5). And secondly, those discussing creative practice aimed at instigating action in the participants themselves, either pre-disaster (D2, F1) for improving preparedness or post-disaster (F3) for improving recovery.

This shows that there is a gap in the academic literature on the use of creative practice with the combined goal of awareness raising and instigating action (middle part of goal axis in Figure 4), and with a broader audience of decision makers and general
public (middle part of audience axis in Figure 4). However, when studying these papers in more detail, we found that they often

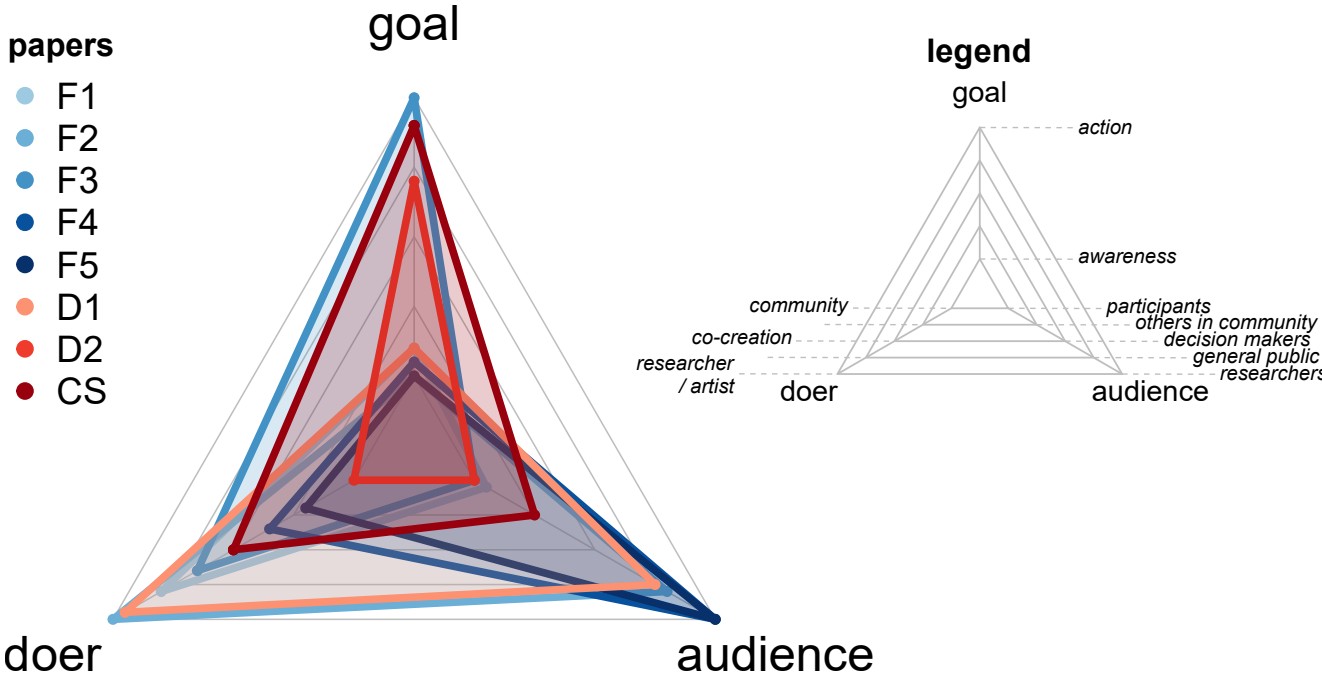

**Figure 4.** Papers on drought and flooding classified in three categories: goal of the creative practice ('goal'), doer of the creative practice / creator of the end-product ('doer'), audience of the creative practice / the end-product ('audience'). Abbreviations and paper details: see Table 1. The goal axis is gradually going from Awareness close to the centre to Action at the top and a combination in between. The doer axis has Community close to the centre, Artist and Researcher at the left-lower end and co-creation in between. The audience axis has five points on the axis: Participants, Others in the community, Decision makers, General public and Researchers. All papers have different colours, the Drought papers and Case Study in red and the Flood papers in blue. The location of the papers on these axes is not accurate but a best estimate.

mention secondary audiences. For example, when researchers were the audience (in F4), indirect impacts on community were noted. And when the community was the audience, insights could 'travel' further to policy makers (in F1, also mentioned in F5), or governmental stakeholders were even included as participants in the creative process (D2). The doer dimension in Figure 4 shows more mixed results between the papers. Compared to the wider literature on environmental issues and health described above, however, these studies on floods and droughts show limited co-creation (Fig. 4; with 5 out of 7 papers having low co-creation, compared to 33% in all reviewed papers), leaving an opportunity to also explore this aspect further. In the next sections, we show an example case study of how this gap might be filled.

All papers assessed here focus only on creative practice as a tool for building resilience to floods and droughts and no combination or comparison with more conventional flood / drought measures (for example structural measures or forecasting

and early-warning) was done. Only F1 and D1 briefly mention the importance of this, with F1 stating that: "more attention needs to be paid to combining structural and nonstructural measures" (Ikeda et al., 2016, p.39). This aspect will be discussed in Section 4.

## 3 Pilot study

The pilot project CreativeDrought (https://creativedrought.wordpress.com/) aimed to develop an approach to local level pre-
paredness to future drought. According to Biggs et al. (2012), important aspects of resilience building are maintaining diversity and redundancy, managing connectivity, managing slow variables and feedbacks, fostering complex adaptive systems thinking, encouraging learning and experimentation, broadening participation, and promoting polycentric governance systems. In this project, we focused on participation, learning and experimentation, and connectivity, and wanted to explore how creative prac- tice could support these. We designed an approach that allowed members of the community to actively engage with potential
futures including sharing of existing local knowledge, experience and strategies ('imagining futures'; Anderson, 2010), and inclusion of scientific information ('calculating futures'; Anderson, 2010). The goal was to create stories about potential future drought impact and preparation / adaptation via a multiple method approach developing text-based narratives, performance, and visuals (video). We used this process to increase dialogue between groups in the community and used the visuals to make the voices of the community members 'travel' to policy makers' circles, where they would not normally be heard. So the
creative practice was jointly conducted between the community and the researchers in an iterative process (doer), aimed at instigating action and raising awareness (goal) by the participants, others in the community and policy makers (audience) (see CS in Table 1 and Fig. 4).

### 3.1 Methods

As case study region we selected a village in Limpopo Province in South Africa. This (anonymous) village was chosen because
of its vulnerability to drought and because the villagers get their water supply from a number of different sources and use it for a range of different purposes, i.e. a groundwater borehole for domestic water supply, two reservoirs for irrigation of agricultural land, and a river for washing, bathing, brickmaking, etc. (Rangecroft et al., 2018). The village has a population of c. 2800 (StatsSA, 2017) and is led by a chief and his royal council. Drinking water supply is organised by the Department of Water and Sanitation and irrigation water is regulated by the Department of Agriculture (Makaya et al., 2020). The village was selected
by our local research partner at the University of Venda and our research assistants were from the village and surrounding area. After a few short initial visits to become acquainted with the area, the community, and its leadership, also building trust and getting permission for the research, the research team spent four periods of one to four weeks over the course of one year working intensively with the community.

We developed and tested an interdisciplinary approach with, as final output, videos of narratives produced by groups of
community members in the village (Rohse et al., in prep.). These were produced in small-scale workshops guided by an interdisciplinary team of researchers and local research assistants, in which the participants were asked to reflect imaginatively

on future drought scenarios produced by a hydrological model. Focusing on narratives as the creative method was a pragmatic choice related to participants' preference for storytelling compared to other (more visual) methods, for example maps or games, and familiarity with these methods by the research team. Additionally, because we focused on imagining of future events, we could not use methods that are rooted in the current or past situation, such as Photovoice.

The development of the narrative videos happened in three phases. In the first phase, we carried out group narrative interviews on the topic of past drought events (Rangecroft et al., 2018; Rohse et al., in prep.). These allowed us to collect rich and contextualised information on past droughts and their impacts on different groups in the community, but also to hear potentially hidden voices and explore the potential for future drought narrative workshops. We used oral history techniques in small group conversations. We did 12 group interviews with 2 to 7 participants each covering a range of different groups within the village (occupation, gender, age). The participants were selected by the village leadership. The interviews were recorded, transcribed, and translated from Venda (the local language) into English by the research assistants.

In the second phase, we set up a hydrological model for the area that could be used to model future scenarios (see Rangecroft et al., 2018). We used the SHETRAN hydrological model (Ewen et al., 2000; Birkinshaw et al., 2010) developed from available datasets and catchment observation. Topography was based on DEM data, precipitation and potential evapotranspiration input came from climate datasets, geology, soil type, land use, and information on location and amount of water abstraction was derived from field observations, dam level and release data, and discussions with knowledgeable locals. The model was qualitatively validated with water level data and with information about the drought events shared by the participants of the narrative interviews in phase 1. The model was run for a baseline run and for three different scenarios (Rangecroft et al., 2018). The scenarios were designed based on conversations with the villagers and with a number of government representatives. The "Warmer Temperatures" scenario was based on an increase of 3°C compared to present day temperatures representing climate change expectations for the region. The "Larger Irrigation Scheme" scenario represents a possible increase in water demand in the future with the area of the irrigation scheme expanded to be twice as large as present day. The "No Dams" scenario was based on the expectation that without maintenance the dams, which were built in the 1960s, might not be available for the community any more in the future. A drought analysis was then performed on the model results for the scenarios and the resulting drought characteristics were compared with the baseline scenario to determine the difference between future and historic droughts (Rangecroft et al., 2018). These results were then translated into storylines for easy communication with the community. These storylines included carefully-phrased information on the expected duration of a lack of rainfall, dryness of the soil in the community plots, and lack of water in the river or irrigation canal (for more details, see Rangecroft et al., 2018).

In the third phase, we organised 6 workshops, in which we brought together different groups in the community, matching the grouping in phase 1 and with some overlap in participants. Again, participants were selected by the village leadership. Our aim was to have around 4 people per group, so 8 per workshop, but due to various reasons attendance was variable between groups. These are the workshops we organised (for more information see Rohse et al., in prep.):

– 2 workshops with orchard farmers and livestock farmers (8 participants in each)

– 2 workshops with young married mothers and elderly women (11 participants and 2 participants)

– 1 workshop with irrigation farmers (older and younger generation) (8 participants)

– 1 workshop with traditional leaders (3 participants)

In these workshops, the participants first listened to the storylines of the possible future droughts, translated and explained by the local research assistants. They then discussed what the impacts of these droughts would be on them and their community and developed these into 'stories' that were filmed. Next, the groups within each workshop exchanged these stories and had a discussion with the whole group about possible responses and preparedness measures. They then went back to their smaller groups to develop this into a story about coping strategies to future drought, which was also filmed. This resulted in two stories about the future for each group, one on future drought impacts and one on future drought preparedness (which took into account the exchange with the other group). The recorded stories were transcribed and translated from Venda into English by the research assistants, allowing us to subtitle the videos.

After the narrative videos were produced, the results of the workshops were discussed in a community forum, shared with community via YouTube, USBs and transcripts, and used in conversations with water management actors. The aim of these conversations was to make marginalised community voices heard to powerful actors and decision makers.

### 3.1.1 Results

The first phase provided very useful information in preparation for the second and third phases. It gave us an understanding of historic drought events and their impacts on the community. The group stories, for example, showed how different groups in society were influenced by different types of drought (livestock farmers by meteorological drought, irrigation farmers by hydrological drought, and domestic water supply by groundwater drought), which all had their different timings and characteristics. The community members, however, did not show understanding of how these types of drought were linked in space and time and their relation with water use (for example, mining activity and a new groundwater borehole for drinking water were not linked to the drying up of springs in the area), but their responses were very helpful in setting up and validating the hydrological model in phase 2. Phase 1 also yielded useful observations for the preparation of the workshops in phase 3. It provided contextual understanding of the challenges the community faced. It highlighted the importance of faith in framing their stance towards possible future drought ("God decides") and difficulties in imagining and talking about the future. We also found that there was seemingly little intergenerational exchange of drought coping strategies, although there were some ambiguous testimonies on this point. And importantly, we tested which creative method would resonate most with the community. We talked about visual methods like artistic maps or other methods like board games for the community to interact with potential future changes in water availability and use. However, from the start it was clear that the idea of 'stories' was most resonant with the community. The participants of phase 1 and the village elders of the royal council indicated their interest in developing stories.

In phase 2, the information collected in phase 1 and during catchment observation was used to set up and test the model. Because the model was used for the development of scenarios, which were then used as starting point for discussion in the workshops, accuracy of model results was not our main aim (Rangecroft et al., 2018). We wanted the model to represent the

past droughts relatively well so that we could trust the modelled potential futures. In phase 1, the community had highlighted droughts in 1983 (mostly mentioned by farmers and elderly men and women), 1992 and 1994-95 (mostly mentioned by young people, married mothers and civic group) and these were reproduced by the model (Rangecroft et al., 2018).

The model was then used to extrapolate and calculate several scenarios that were mentioned by community members and government representatives. Instead of predicting the future, we were exploring plausible futures. Droughts were calculated and compared between the scenario and the baseline. These were transformed into storylines including information on the duration of future droughts and a qualitative indication of severity (e.g. more severe than has been experienced in the past 40 years). Figure 5 shows the process of creating and communicating the storylines to the workshop participants. We used one climate change scenario and two scenarios related to human activities (i.e. increased water use for irrigation and decreased water availability due to lack of dam maintenance), but found that the latter were more difficult to communicate, possibly due to the limited knowledge of the relationship between water use and water availability (as mentioned before) or because there were some political sensitivities related to water use by neighbouring communities, so in the end, we used the climate change scenario in most of the phase 3 workshops.

The workshops generated very rich information on potential future drought impacts and possible coping and preparedness strategies. These did not necessarily develop into fully-formed stories, but they did clearly communicate strong emotions and allowed for imagining positive options. Compared to phase 1, in which narrative-style answers were also used but where people found it hard to engage with uncertain futures, in the workshops the narrative approach supported by data from the model scenarios allowed participants to use their imagination and exchange ideas. Some participants now made the link between water use for irrigation and water available for domestic purposes, which did not happen in the phase 1 group interviews (Rohse et al., in prep.). We found that the intergenerational exchange was very powerful, with older farmers willing to share traditional techniques and younger farmers eager to learn. The exchange between participants with different professions also worked well, although there was already an awareness of the needs of different groups in the community, mostly because these are not strictly separated, with for example livestock farmers also having a small plot to grow vegetables and being domestic water users as well. We also found that there were very different preparedness and coping strategies brought forward in the different workshops, including proposals for individual actions (digging for water in the river bed, selling livestock, saving food), community actions (collective maintenance of the irrigation system, drilling a new borehole), and requests for government support (providing food, fodder, drinking water, and jobs).

The approach allowed participants to use their imagination and consider future drought events, their impacts and preparation, and to exchange ideas between different intergenerational groups and across different professional occupations. The research assistants who carried out the bulk of the facilitation in the local language, reported that whilst some participants were a bit concerned with how long the workshops were, there was general enthusiasm for the topics discussed and participants had many ideas to contribute, and valued the opportunity to have a platform to exchange and learn from other community members. For example, younger people were eager to learn from older people about the traditional methods for community and household resilience (e.g. food storage techniques) that had largely fallen out of use.

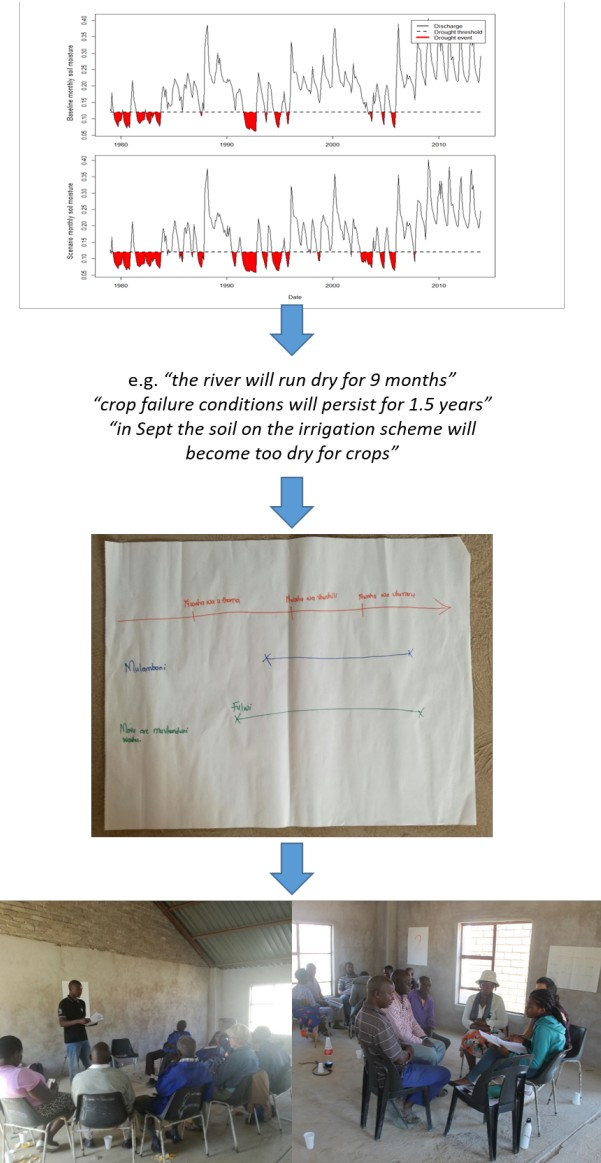

**Figure 5.** The process of translating model scenario results into storylines in the local language that were then explained to the workshop participants by the research assistants (photos by Sally Rangecroft).

The recorded and edited narratives (Fig. 6) were given back to the community with the idea that these could be used in schools and community gatherings. The narratives were also a useful tool for creating space for conversations with government representatives about local perspectives on drought management. Whilst initially, the local policy makers we followed up with were more interested in the model data than in the community narratives that were developed from the model results, after carefully facilitated discussion they acknowledged the value of the videos in sharing the concerns of the community. The

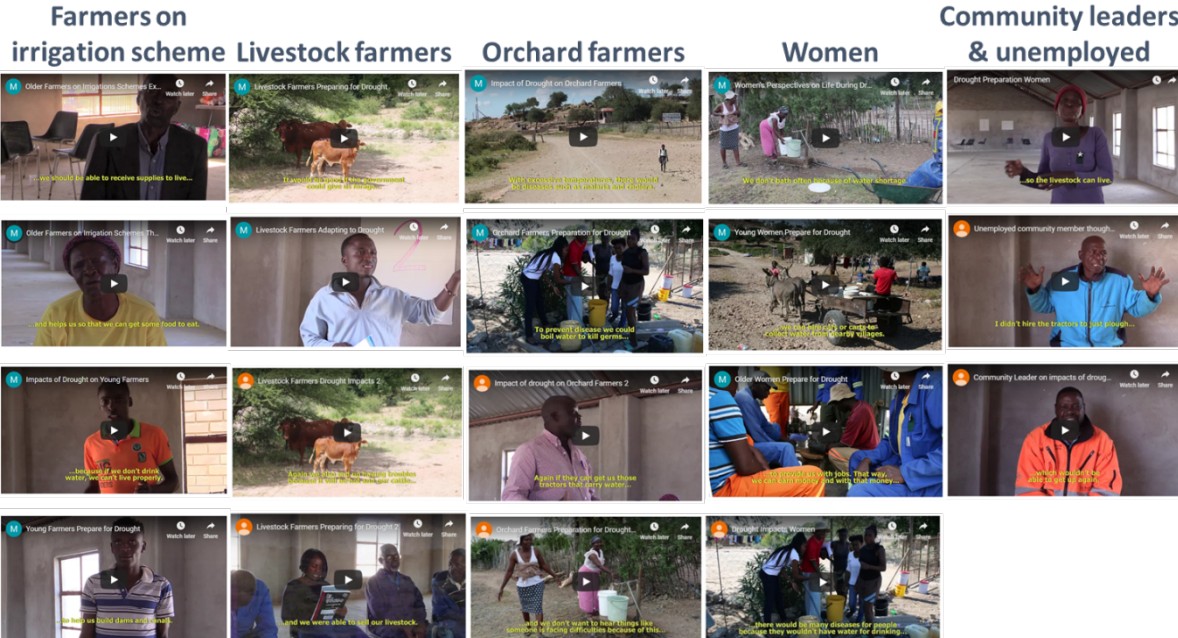

**Figure 6.** Recorded and edited future drought narratives developed in community workshops in the CreativeDrought project (see https://creativedrought.wordpress.com/videos/).

videos proved to be stimulating prompts for conversations on drought preparedness and on the current situation facing the community. In addition, the policy makers found some of the images useful as illustrations of the specific conditions facing the villagers, as they explained it is sometimes difficult to get reliable information on villagers' situations. It was clear that they struggled with their role balancing between supporting the community and empowering them to face drought challenges without relying too much on government support, reflecting some of the tensions in the notion of resilience that we referred to in Section 1.

## 4 Reflections & perspectives

We identified a clear gap in the academic literature on using creative practice to build resilience to hazards with a focus on Global South communities. The seven studies on floods and droughts we analysed (Figure 4) either used creative practice for raising awareness of the general public or researchers (by using it as research tool) or for instigating action by the participants. Although there was some mention of secondary audiences of the creative practice (for example others in the community or the general public) and of how end-products could be used in decision making, these aspects were not explored further. Also, the creative practice was mostly either done by the community or by the researcher team and co-creation was limited. The other 260 studies with a wider focus selected in our literature mapping exercise (encompassing other environmental issues and

disasters, climate change, health, socio-economic inequalities, and violence & conflict) showed a similar pattern with most art-based climate change communication focused on raising awareness of the general public with little co-creation, and most creative practice in the field of health focused on encouraging behavioural change of participants with high co-creation.

This gap is what we aimed to explore with our pilot study. In an iterative process, we developed stories with the community with the aim to instigate action to build resilience to future drought, both by the participants and others in community. By filming and editing the stories developed by the community, we were able to use them as a discussion starter with decision makers and bring some of the community voices to more powerful actors. In this way the products of the creative practice also had the goal of raising awareness. Unfortunately, due to the short nature of the pilot project funding, we were not able to evaluate

the effectiveness of communicating via stories compared to more established ways of communicating and to investigate how these art-based ways of building resilience can be combined with more conventional ways of dealing with drought. This is a common feature among much of the research on art and creativity in environmental and health-related issues. Most papers we reviewed described a methodology and promoting a potential beneficial method without clear evaluation of its efficacy. In this section, we discuss the limitations of our study and share our reflections and thoughts on the ways forward.

**4.1  Limitations**

The results of both our literature study (reported in Section 2) and our pilot study (Section 3) should be seen in the light of possible limitations. The limitations of our literature study include:

- focus on academic literature: we only mapped academic papers and did not include the wealth of creative practice for resilience building used by artists, NGOs and other organisations. These projects are often shared via reports and on

websites and video platforms and we found that searching these led to a strong bias related to the keywords and platforms used.

- language bias: we only investigated papers published in English. This leaves out much published research on the topic of creative practice and environmental issues in other languages. In particular, Latin America featured less in our literature review than expected (Figure 2), because research from that region is mostly published in Spanish or Portuguese (e.g.,

Gomide et al., 2019). The same language bias is visible in other review papers on this subject (Nurmis, 2016; Galafassi et al., 2018; Coemans and Hannes, 2017; Teti et al., 2018; Fantini, 2017).

    The potential limitations of our pilot study are mostly related to the short duration of the project, which was funded for only 1.5 years. This led to the following issues:

- We were not as embedded in the community as we would have liked, which meant that we were dependent on the village

leadership and our local research assistants for selecting and communicating with participants.

- Interaction with policy makers was limited to a few exchanges at the start and end of the project. Although government representatives showed interest in testing our approach in other communities, we did not have the time in the project to

embed our creative practice in the decision making process and combine it with more conventional measures for dealing with droughts, which were more familiar to the decision makers.

– Within the timeframe of the project, we could not evaluate the long-term benefits of our approach.

## 4.2 Reflections

At the start of this paper, we asked the question whether creative methods can support preparedness to different types of hazards. Based on our results there is no clear answer to this question, mostly because there is a lack of evaluation of the effects of approaches using creative practice. For example, it is as yet unclear whether narrative workshops, like those used in our

pilot study (Rangecroft et al., 2018; Rohse et al., in prep.), are more effective than more conventional workshops, as used by Ikeda et al. (2016). Only in (mental) health research where art-based methods are used therapeutically there is some evidence for their effectiveness (e.g. Stuckey and Nobel, 2010; Van Lith et al., 2013; Slayton et al., 2010). Photovoice (participatory methods using photography) has also been found to have tangible effects on social justice, albeit mostly on raising awareness and causing only limited transformation (Sanon et al., 2014). One of the issues is that this transformation often only happens

on longer timescales, beyond the lifetime of many research projects.

Many of the papers we reviewed in our literature review did mention short-term benefits of the creative practice (e.g. Contreras et al., 2018) and also in our pilot study we noticed some positive effects of our methodology. Feedback from participants can be used to give an indication of these short-term effects. For example, Strickert and Bradford (2015) reported that participants of a theater performance found the experience 'interesting, legitimate, and effective' for learning about water management

challenges. In our pilot study, both younger and older participants enjoyed the intergenerational exchanges that our approach encouraged. This was in stark contrast with the phase 1 interviews, in which we often heard the complaint from the older generation that young people did not want to listen to their stories. Also most local decision makers indicated that they found the community stories we captured interesting and useful.

When evaluating the efficacy of creative practice for building resilience to hazards, the timescales of potential effects need

to be taken into account. Creative practice with the goal of instigating action will mostly have short-term and tangible effects, in our pilot study for example fixing leakage in irrigation channels, not building structures in the floodplain, storing food. Creative practice with the goal of raising awareness could have more long-term and intangible effects, for example a change in beliefs or values. However, both might have long-lasting side-effects such as changed interactions within the community or between the community and government. It is on these longer timescales that creative practice could be a catalyst of deeper

transformations. Artistic products are often enduring in their relevance and might be used long after they were developed, but a question is whether benefits are mostly coming from the creative process (benefiting the doer) or from the product (benefiting the audience). This determines how long-lasting the effects of engaging in artistic practice are for building resilience against natural hazards. To evaluate these long-term outcomes, longer research projects are needed.

Some scholars have critiqued the use of art-based methods, for example by noting that art distances the problem (Miles,

2010) or by questioning whether art-based methods can actually achieve any change (see Nurmis, 2016). Apocalyptic climate

change art, for example, can lead to fatalistic views that do the opposite of instigating action and behavioural change (Nurmis, 2016). In our pilot study, this was not the case, rather the reverse happened: we noticed a first fatalistic reaction to the future scenario storylines, but the further engagement, exchange and narrative development actually transformed this fatalism into ideas and suggestions for adaptation measures.

Based on our studies, we see a large potential for using art-based methods. In the introduction, we mentioned that preparing for future extremes requires including diverse knowledges, elevating under-represented voices, thinking out of the box for possible solutions, enhancing communication between diverse groups, and instigating organisational and behavioural change. For some of these elements, creative practice seems to be useful. In our pilot study, we could see effects of thinking out of the box, enhancing communication, and elevating under-represented voices. For example, we saw a clear difference between

the focus group interviews in phase 1 and the narrative workshops in phase 3, with participants in phase 3 showing more imagination of potential futures and how adaptive measures could help, and more exchange between groups in community. Furthermore, the community message was conveyed to policy makers in a way that was unfamiliar but interesting to them and they noted that they got a better picture of the challenges the community were facing. Creativity can also foster exchanges in space and time. For example, a community that has not yet experienced a drought and may be confronted by these events in

the future (for example due to climate change) could get an idea of what it is like on the ground by learning from communities facing droughts on a regular basis. This increased understanding of the challenges and possible preparedness measures could help them to be better prepared.

    Both in the reviewed literature and in our case study, we encountered many barriers and practical challenges to using creative practice in resilience-related research in the Global South. These practical challenges include:

– language: in cases where the researcher does not speak the local language or is not from the area being researched, knowing the full breadth of traditional art-based methods or translating the details of creative practice activities and intended outcomes can be challenging. However, language might be less of a problem in some non-verbal art and creativity than when using other more verbal methodologies, such as interviews or archival research.

    – time: creative methods often take a lot of time, both from the participants and from the researchers. Researchers need to

gain in-depth insights of their case studies and if translators are involved (see language), they need to be well-briefed. Also, if art-forms are used that the community is not familiar with, a thorough explanation is needed.

    – unfamiliarity: participants are not always comfortable in engaging in creative practice and some decision makers tend to prefer quantitative outputs with a specified (un)certainty instead of contextualised stories, photographs, songs or other art products (Owens, 2005).

To overcome these challenges, good facilitation is crucial. Ideally, the research team is interdisciplinary and including local researchers and the work is guided and facilitated by creative practitioners / artists speaking the local language and familiar with participatory art-based research (see e.g. 'social volcanology', Donovan, 2010).

    Just like it is important for more conventional structural solutions to floods and droughts to be adapted to the local climate and land surface conditions to be effective, it is important for non-structural solutions also to be embedded in local circumstances

(both natural, socio-economic and cultural). With art-based methods, the latter could potentially be done more easily when art forms are used that are deeply rooted within the culture of the community. This would potentially ensure the longevity and effectiveness of the intervention. In our pilot study, participants preferred storytelling as it is an art form already rooted in their culture. This also means that creative practice methods and art forms might not be transferable across communities as different communities have different histories / sensitivities to different creative practice. Also policy makers have their own preferences for certain types of evidence in policy making (e.g. Jasanoff, 2013). In our interaction with policy makers, we noticed more familiarity with and interest in more quantitative types of information (model results) instead of qualitative ones (stories).

It is important to stress that both types of information are still needed for better preparedness for future hazards. Structural measures, improved early-warning systems, all of that is indispensable for reducing socio-economic impacts of hazards and loss of life. However, without acknowledging local knowledge, circumstances, and challenges, and without empowering local communities, changing their behaviour, encouraging uptake of both structural and non-structural solutions, these more conventional measures might not be used or not used effectively. It is widely recognised that improvements in flood and drought resilience need to come both from communities and government as a shared responsibility between public and private partners (Trim, 2004; Amaratunga et al., 2009). However, Global South communities often face many challenges and barriers for actions, including lack of resources (land and financial resources), unemployment, and lack of information (such as drought early-warning) and creative practice on its own is unlikely to be able to solve these. In our pilot study, for example, it emerged that for the severe future drought scenarios (outside previous experiences) community members suggested that bigger infrastructural changed were needed (either done as community or provided by the government). However, if the creative practice is part of a larger programme that integrates local and scientific knowledge and combines community-focused activities with activities aimed at decision makers, it may have greater impact potential.

McMillen et al. (2016) showed, based on an example from Hawai'i, that arts-based approaches to community resilience have alternating phases of being more and less important, in relation to socio-ecological shifts over time. We feel that globally we are now living in a time that asks for more creativity in relation to environmental issues and disasters. Traditional methods for natural resource management are either forgotten (Janif et al., 2016) or inadequate in the Anthropocene (McDowell and Hess, 2012; Kareiva and Fuller, 2016; Head, 2016). Adaptation based on experience might have worked in the past. For example, the process of sharing oral history accounts can contribute to community resilience (Osterhoudt, 2018), but how does it apply to future events? There is evidence that damage from natural hazards decreases after repeated events due to adaptation (for floods: Kreibich et al., 2017). Memory of a previous event has been found to decrease damage in the next event (Viglione et al., 2014), but this memory decreases over time (Lopez et al., 2017). A big question is how to increase resilience if the previous extreme event was very long ago or when future events are outside of previous experience? Alternative ways are needed that require out-of-the-box thinking to imagine how the future might be different from the past and what actions are needed to prepare for this future. Creative methods might be able to enhance this process, especially if they are embedded in or making use of traditional ecological knowledge (e.g. Laidler, 2006).

## 4.3 Suggestions for future work

For future research on using creative practice to build resilience to future drought and flooding, we suggest to focus on the following aspects:

- Firstly, we believe we can learn more from cultures and times where / when art and creativity were used for coping with environmental issues. Indigenous knowledge systems have had to deal with climatic and anthropogenic change (McMillen et al., 2016; Gibson and Gordon, 2018) and might show ways to use art and creativity for natural resource management under changing conditions (e.g. Berkes et al., 2000; Whyte, 2018). An interesting example is Aboriginal use of art for connecting to the land in Australia (Rigby et al., 2011; Zurba and Berkes, 2013). More knowledge on traditional ways of using art and creativity for building resilience could support contemporary initiatives.

- Secondly, research is needed on evaluating the three dimensions of creative practice (goal, doer, audience; see Sect. 1). Is the aim to communicate information or awareness, or to instigate action? And who needs to have more awareness or take action: the participants of the creative practice or others, such as the general public, researchers or decision makers? It is especially important to evaluate whether the methodology of the creative practice fits the intended outcomes.

- Thirdly, we want to encourage more research on co-creation during the creative process. How can artists and researchers work together with local communities to ensure mutual learning? Can / should decision makers be included in the creative process and not only be receivers of the end-product? We feel that the use of participatory art is currently underexplored. As Rathwell and Armitage (2016) noted: "collaborative art making is a platform for knowledge coproduction, whereby novel ideas or products emerge from different ways of knowing".

- Fourthly, more evidence is needed on the effectiveness of creative methods. We call for empirical research comparing creative methods to more conventional methods of doing research, communicating information, or instigating action in relation to droughts and flood. Research is needed on the drivers of any observed change: is the reason for change that the members of a community are coming together and exchanging ideas, or is it the creative practice allowing participants to think outside the box? One of the methodological challenges in evaluating effects and drivers is that change can happen many months after an intervention. Longitudinal studies (i.e. doing research in the same community over a longer period of time) and being embedded in the communities would be needed (Donovan, 2010). Also comparative studies might shed some light on this issue, for example between coping with drought in aboriginal communities where the use of art seems to be deeply rooted in culture vs. in a culture where art is very rarely used.

- And finally, we suggest research to investigate how creative practice can be embedded in a holistic strategy for building resilience. Can creative practice support communities in Global South settings to communicate their worries or traditional methods of dealing with environmental issues more effectively? How can art-based non-structural methods be combined with more conventional structural measures to mitigate drought and flooding? Like Whittle et al. (2012), we argue that creative side of resilience cannot be separated from the physical and socio-economic aspects of resilience.

## 5 Conclusions

In this study, we investigated how creative and art-based methods can support a transition to more resilience to natural hazards, and more specifically droughts and floods, in Global South communities. We mapped the existing literature on this topic (Sect. 2), presented a pilot case study (Sect. 3), and shared the limitations of our work, our reflections, and perspectives for future research (Sect. 4). Our literature mapping and case study indicated that there is a potential for artistic and creative methods to be used more for building resilience to drought and flooding, complimenting more conventional methodologies.

Currently, research focusing specifically on creative practice for flood / drought resilience is limited, although there is a wealth of research on using creative practice in fields of health, environmental issues, and climate change communication. Within the literature there is scope to explore more on creative co-creation of artistic products and on audiences beyond the participants themselves and the general public. Several advantages of using creative practice emerge from our literature mapping and pilot study: it can surface hidden voices, communicate issues on a deeper, more emotional level, travel to wider audiences, increase engagement and behavioural change. Potential challenges are language barriers between researcher and participants, time investment of both researchers and participants, and possible unfamiliarity of participants with artistic or creative methods or products.

Whether creative practice leads to action improving resilience to droughts and floods in marginalised communities is an unresolved question. An important reason for this is the long-term and intangible effects of these types of interventions, which are rarely evaluated. Feedback from participants indicates a number of short-term benefits of creative practice approaches, including more understanding of the issues, increased interaction within the community, and less fatalistic, more positive ideas for adapting to future drought.

Based on the literature mapping and pilot study we call for more research on the use of creative practice in building resilience to extreme events. It is especially important to investigate how the use of creative methods compares to other methods, and how effective creative practice is at bringing about change, either in people's behaviour or in measures implemented by decision makers. We also think future research should address the question how to combining creative methods with more conventional scientific methods and decision making. For decision makers a combination of quantitative, qualitative, and creative information might be most successful in supporting marginalised communities in coping with drought and flooding.

*Data availability.* The literature review database (Sect. 2 and Appendix A) will be made openly available upon publication of the article.

*Video supplement.* The narrative videos developed in the pilot study (Sect. 3) are available here: https://creativedrought.wordpress.com/videos/

## Appendix A: Methodology literature review

In the literature mapping exercise, we reviewed peer-reviewed journal articles. We searched the databases of Science Direct, Web of Knowledge, JSTOR, Taylor and Francis Online, ProQuest, Academia.edu, Project MUSE, and Dimensions. We limited our search to the period 2000-2018, as in previous reviews on related topics art-based research has been found to be limited before 2000 (Coemans and Hannes, 2017; Galafassi et al., 2018). When searching these databases we used a combination of keywords describing different sectors, different art and creative forms, specific regions in the Global South, and words like 'participatory', 'indigenous', 'community'. The search process was iterative, with search terms adapted when they did not yield the expected results. Like Coemans and Hannes (2017), we found that searching by specific art type works better than using 'art' in general. This yielded many papers on the 'state of the art' in certain research field. The same holds for geographic area: most researchers do not use the keyword 'Global South' in their titles or keywords, but rather mention the specified region or country/countries. We focused our search on Low- and Middle-Income Countries and on vulnerable groups within High-Income Countries (indigenous groups, refugees / asylum seekers, children / young people). We limited the search to papers in which art was used as research process and art was developed as product from the research. Articles on art therapy and projects using art therapeutically were excluded. This required a manual and iterative search process of removing duplicates and irrelevant articles by screening titles and reading abstracts. We then further explored the articles using a descriptive coding scheme to filter out information (aim, art form, degree of co-creation). This coding scheme was designed based on the first few papers and then refined during the analysis. For art form, a range of detailed categories were used that were later merged into the main categories: photography, music & song, visual art, drama, storytelling, video & film, dance, craft, poetry, new media, and body modification. For degree of co-creation, we used a qualitative distinction between low, medium and high co-creation. Data were extracted from the abstract and rest of the paper if needed. The papers on the topics of drought and flooding were classified into a matrix based on the following categories: goal, doer, audience. Finally, the papers were summarised to easily extract information.

*Author contributions.* AVL conceived the study with input from MR, PJ, and RD. AVL, PJ, and RD designed the literature mapping exercise and ILM carried it out. MR designed and carried out the community workshops in the pilot study, supported by AVL and RD. AVL prepared the manuscript with contributions from all co-authors.

*Competing interests.* The authors declare no competing interests.

*Acknowledgements.* We thank our funding sources for this study: the Institute for Global Innovation of the University of Birmingham for funding the literature mapping exercise and NERC-ESRC-AHRC via the GCRF Building Resilience call for funding the CreativeDrought project (grant number: NE/P016049/1). We also want to thank our local partners and co-facilitators in South Africa for helping with the

pilot study (Professor Edward Nesamvuni, Livhuwani Ludick Khobo, Tshimangadzo Mandoma, Ndivhuwo Makhalimela, and Khutadzo Ndwambi) and are indebted to the chief and the community for welcoming us into the community. We thank the conveners of the European Geoscience Union (EGU) 'Scientists, artists and the Earth: co-operating for a better planet' for inviting us to present this work at the EGU conference and in this Special Issue. And thanks to the reviewers Louise Arnal, Susanne Maciel, Zareen Bharucha, Mathew Stiller-Reeve and colleagues for detailed feedback on the draft version of this manuscript. This research contributes to the IAHS Panta Rhei initiative, and specifically the working group on Drought in the Anthropocene. Finally, we want to thank the research group at IVM-VU Amsterdam for helpful suggestions for the figures.

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
