# Peer review of "Creative practice as a potential tool to build drought and flood resilience in the Global South"

_Geoscience Communication, 2020_

## Short Comment (SC1) · 20 Mar 2020

This is a brief log of minor errors I've spotted in this version of the manuscript, and a list of a couple of opportunities for further clarification, if these would be helpful.

1. Change around the order of a and b in the references in these sentences: "Global South communities are vulnerable to the impacts of floods and droughts, and are expected to be even more at risk in the future (Winsemius et al., 2015b), as increased climate variability and is likely to lead to more floods and droughts (IPCC, 2012) and water demands and exposure and vulnerability are growing (Wanders and Wada, 2015; Winsemius et al., 2015a)."

2. Line 30: Can you give a couple of examples of what you mean by creative practice

[Figure]

/ processes? At the moment the sentence is also reading a bit circular-ly (if that's a word!) Also might be worth giving an example of artistic artefacts to better illustrate this difference.

3. I'm not familiar with this literature, but in Lines 33 to 40 where you are defining Global South communities would it be worth specifying whether you are meaning marginalised communities (i.e. socioeconomically disadvantaged, spatially distinct 'communities of place', wherever that place might be) or 'rural communities' (which may not all be marginalised - I'm typing this on the train outside Newmarket ;)

4. Line 41 - Maybe in one sentence or even within this one, specify what these critiques are saying (e.g. lack of attention to power relations / diverse knowledges?) Also may be worth expanding a bit on how you're defining resilience?

5. Line 45 - complex interactions

6. Line 76: 'the' audience'?

7. Line 77: 'a' therapeutic way?

8. Lines 170 onwards: I think this is fascinating..so... there is a lot of work beginning to be done, and momentum is building, but we're not sure yet if these methods are really shifting things for people on the ground? A sniff a potential long-term research project, or something looking at long-term outcomes. What changed for participants, how significant was the art practice as a catalyst?

9. BUT having said this in the discussion you touch on there being a lack of effort to do practice that may be used in decision making. Maybe there's a discussion point around the kinds of impact that might be considered worthwhile - things that may have a tangible, immediately discernible impact may not be the ones that catalyse deeper 'transformations' in values for example, and maybe different kinds of interventions are needed for these different 'levels' of change? I'm thinking here of Marina's PhD work (which you heard a bit about yesterday), and her absolute insistence on not claiming

particular kinds and directions of change, and 'art' being a relatively unpredictable 'space'. But then her practice is entirely different.

10. Table 1 - what a great resource! This is really helpful for others in the field.

11. Line 190: A space is missing in the first word

12. Line 192: "into in" should be 'into'

13. Line 270: 'It gave us an understanding...'?

14. Line 304: Aha - this is super, and I wonder if it's worth (later in the paper) making it a key point of discussion - there seems to be something there about the practice re-locating agency?

15. Line 328: (Re)specify what this gap is and why it matters?

---

## Referee Comment (RC1) · Louise Arnal (Referee) · 3 Apr 2020

I really enjoyed reading this paper, highlighting gaps in the literature about the use of creative practice to foster awareness and promote action and collaboration in marginalised and vulnerable communities facing environmental issues. It presents a fascinating pilot project with a community in S. Africa, where storytelling workshops were organised to create narratives about impacts of and preparedness for future droughts. This tackles some of the gaps previously identified in the literature, and complements a still mostly unexplored research field into the use of creative practice to increase resilience to natural hazards. In my opinion this paper is more than suitable for publication in Geoscience Communication, for the special issue's theme of 'Five years of Earth sciences and art at the EGU (2015–2019)'. In particular, I really enjoyed

reading about the current literature in the field and discovering the pilot project (an amazing project - congratulations!) and how it aimed to tackle some of the literature gaps (which are very clearly defined). I however have a couple of minor comments that I would like to raise, and which I would encourage the editor and the author to consider prior to publication:

Main comments

- The authors seem to suggest throughout the paper that the impacts of using creative practice should be analysed against using more traditional methodologies. E.g. on P1 L16-17 and P16 L398-399. In my opinion however, and as raised by the authors in the last bullet point on P16, creative practice should be used in combination with more traditional methodologies. As such, the aim should not be to compare the impact of both processes, but rather to investigate the added value of creative practices within more traditional current processes.

- As a succession to this point, it would be great if you could strengthen your point on how creative practice can complement more traditional methodologies, perhaps in the discussion. This is very nicely reflected in the methodology you follow for the pilot project, as explained on P10 L228-229: the use of model outputs to prompt the participants' imagination. A few questions it would be great to have your opinions on are: 1) How can creative practice tackle the point you raise on P1-2 L23-26? E.g. you mention on P15-16 L375-379 that creative methods can help increase resilience if previous extreme events happened a long time ago or for future events outside of previous experience (which is expected to happen more frequently with climate change). 2) Another point is that the success of increasing dialogues between groups in a community is better measured on the long-term (P10 L216-217). Could you please reflect on how creative practice has a role in insuring this "longevity" of the success of a project. E.g. I would argue that art is timeless, whereas other more traditional methodologies might not be. 3) As hinted by your comment on P11 L230, different art forms appeal to different communities given their culture. E.g. some communities might prefer sto-

rytelling, others dancing, etc. as these are art forms already deeply rooted within their culture. This is where creative practice can help over more traditional methodologies, by echoing a community's culture.

- I found it interesting to read about the different examples of creative practices and their goal, doer and audience on P3-4 L70-110, but found it hard to understand the exact purpose of these paragraphs. It appears to already be a part of the literature mapping and hence might fit better in the next sub-section? If their purpose is to give examples of different combinations of the three dimensions you put forward earlier, it would be helpful to clarify this. It might also help guide the reader to be consistent and use the same language as introduced earlier regarding the three dimensions within these five example paragraphs. E.g. For the first point, the goal is to raise awareness by passing on knowledge between generations, the doer and audience are the community.

- Where do games stand in the midst of the creative processes you looked at? I would argue that they are a creative practice. However, there is very little mention of games until P11 L230 (if I'm not mistaken). On P4 L109-110 you say that there is no example to your knowledge of artistic products in decision-making. I think that there are plenty of resources on the use of games in decision-making which it would be great to highlight. E.g. the numerous workshops organised by the Red Cross using "serious games": https://www.climatecentre.org/resources-games/games, the IHE Delft games: https://www.un-ihe.org/serious-games-decision-making, or the HEPEX games: https://hepex.inrae.fr/resources/hepex-games/.

- I found it hard to understand Fig. 4 and visualise the results you mention on P9-10 L196-203. I think another graphic format may be more suitable to highlight these results and the gaps in the existing literature. Pie charts may be more intuitive? Could you please also change "Method" to "Doer", to be consistent with language introduced on P3 L63. It is not clear to me why 2 of the CS and one F5 circles are lighter in colour. Additionally, I think it would be very interesting to be able to map the wider literature onto this graphic to see how the flood- & drought-related literature compares to it.

- Could you please reflect in the "reflections & perspectives" on: 1) how your findings may be limited by the mapping methodology you used: searching for research papers. There are surely a lot more creative practice examples out there, less research-oriented and with different goals, audience and doer, but not mentioned in any research paper. 2) What worked very well and less well in your pilot project: would you do anything differently now? What tips would you give to people who want to create such projects? 3) How do you foresee the continuation of the project? Do you plan to put in place methods to evaluate the long-term impacts this project may have on the community and decision-makers?

Specific comments

- P1 L20: Could you please define here what you mean with Global South communities, perhaps by moving the definition on P2 L33-34 here.

- P2 L25-28: These 2 sentences seem like a repetition. Please considering merging and/or reformulating.

- P2 L31-32: This statement puzzled me at first. Could you maybe give examples of creative practice (with and without end products) and of artistic artefacts.

- P2 L37: Could you please clarify what you mean by "suitable traditional structural or non-structural measures".

- P2 L41: Could you please explain briefly what the main critiques of the term "resilience" are.

-P3 L63-64: The examples of goals you mention here do not correspond to the goals you mention on P4 L112. Please consider updating this list, as well as throughout the paper.

- P3 L73-74: Please specify what is the purpose of the practice described by McEwen et al.

- P4 L93: I would argue that here the audience would also be the end-users of the research product, for example the readers of the research paper if the research is published, who may or may not be researchers.

- P4 L106: Could you please clarify what you mean by "mental models".

- Fig. 1: If you have the data to plot this, it would be great to be able to visualise the separation per wider topics as well (i.e. hazards and disasters, climate change, other environmental issues, health, social and economic inequality, violence and conflict; or broader topics even), as different colours/patterns on the bars, to see how these change over time. E.g. Are the natural hazard articles more recent, even if they make a small overall portion of all papers?

- P6 L 140-141: Could you please clarify what is "Photovoice (or Photo-Elicitation Methods or Camera-User-Study)" for less familiar readers like me.

- P6 L 145-146: I understood this sentence only after having finished reading the whole paragraph. Could you please rephrase to clarify what is meant by "asking participants to develop new material".

- P7 L166-167: Would you be able to give an estimate of the % for all environmental papers you looked at in the literature mapping? It would be nice to have it to compare the % with those for papers on droughts & floods later on, on P10 L200-201.

- P7 L167: Could you please clarify what you mean by "medium or high" (also on P8 L169).

- P8 L176: Please clarify that these papers are D1&2.

- P8 L184-185: This is a repetition of the line on P8 L180-182. Please consider merging.

- P8 L186: Please clarify that these papers are F1-5.

- P9 L192: Please specify here that this is referring to Fig. 4.

- P9-10 L196-199: This is not clear to me. Are the 2 categories you mention within the "goal" and "audience" categories? If so, could you please rephrase these sentences. The choice of the words "Firstly" and "Secondly" may be confusing me. Could you please also discuss what results are with regards to the "Doer" (or "Method").

- P10 L199: Is "instigating action" pre-disaster similar to "raising awareness"?

- P10 L201: It would be great if you could mention again here what the percentages are of studies with a co-creation aspect, for comparison, for: floods & droughts, health issues and environmental issues.

- P10 L210: Could you please clarify what you mean by "experimentation".

- P10 L212: You use the plural form of "approach" on P10 L207. Is this because the project was made of several approaches which your creative approach was a part of? Please clarify in the text.

- P11 L230: This is very interesting and merits further discussion. It appears that participants preferred storytelling as it is an art form already rooted in their culture perhaps? It would be great if you could add a few lines in the discussion about how processes/media might not be transferrable across communities as different communities have different histories/sensitivities to different arts. This perhaps complements your second point on P16: the goal, doer and audience are situation-specific, but so is the creative form/media used.

- P11 L240: Do you have any reference you could cite here for the SHETRAN model?

- P11 L252-253: I imagine the phrasing of these scenarios was worked on with care as they could lead to different responses from the participants. Could you please say a bit more on how the storylines were written? And maybe give a written example of a model output and the attached storyline in the paper?

- P11 L254: Are the workshop categories in Phase 3 the same as in Phase 1?

- P12 L287-289: Do you think this is just because of the age difference of these groups? These sentences would fit better in the former paragraph I think.

- P13 L296-297: Please specify that you are referring for the workshops of phase 3.

- P13 L311: Do you mean "request for government support"?

- Fig. 5: It would be great, if possible, to have tangible narrative content in the paper as well. Only if possible, please consider adding perhaps parts of a narrative transcript, or a few quotes from several narratives, to the paper.

- P14 L329-330: Could you please share opinions on why you think that is?

- P14 L333-335: I am not sure I understand this sentence. Could you please clarify.

- P14 L336-340: I think you are being too harsh with yourselves. While you couldn't, as part of this project, evaluate the long-term benefits of your creative practice, there are some clear short-term benefits that are worth mentioning again in the discussion. E.g. the fact that policymakers found the images you captured useful, the intergenerational exchanges your workshops led to, the fact that your workshops encouraged participants to use their imagination and exchange ideas vs referring to God in an uncertain future. . . These are already brilliant outputs which should feature here.

- P15 L371-373: This feels like an important point. Could you maybe answer this question using your pilot study? As you have seen both aspects in phase 1 and 3 of your workshops.

- P15-16 L375-379: This is where creativity can foster exchanges across communities. E.g. A community who has not yet experienced a drought and may be confronted by these events in the future (for example due to climate change) could get an idea of what it is like on the ground by learning from communities facing droughts on a regular basis.

- P16 L382-386: Could you please explain the purpose of this point further.

- P16 L404: Could you please clarify what "longitudinal studies" are.

- P16 L396-405: Could the effectiveness of creative practice be measured by comparing different communities and how they have coped with environmental issues? For example aboriginal communities where the use of art seems to be deeply rooted in culture vs a culture where art is very rarely used.

-P17: Please summarise briefly what your paper is about before mentioning results.

Technical corrections

- P1 L21: Remove "and".

- P1 L22: Replace "and" after "water demands" by a comma.

- P3 L73: Add a dot after "environmental stress".

- P3 L73: "describe" without an s.

- P4 L89: "built on" with a t.

- P4 L111: Add "we" before "will".

- P4 L112: "instigating".

- P7 L156: "developed".

- P10 L209: "redundancy" might not be the adequate wording. Do you mean "repetition"?

- P10 L217: "members' travel to".

- P11 L247: "3°C".

- P11 L247: Remove "a" in front of "climate".

- P12 L294-295: "We used one climate change scenario and two scenarios related to human activities".

- P14 L323: Add "the" in front of "images".

- P14 L324: "faced by the villagers"?

- P17 L422: "combine".

---

## Author Comment (AC1) · 22 Apr 2020

We want to thank Zareen Bharucha for commenting on our manuscript. In the next version of the paper we will address her comments. We will fix the textual errors and we will add some example of what we mean with creative practice and artistic artefacts. Here some initial answers to the questions raised in SC1.

3. We mean marginalised communities, not (necessarily) rural communities. This encompasses vulnerable and socio-economically disadvantaged groups in society, which are more abundant in the South, amongst ethnic minorities in both South and North, and amongst more rural populations, but not exclusively. There might be marginalised communities in big cities and some rural populations might be very wealthy and well-

protected against natural hazards. We will clarify this distinction in the manuscript.

4. On line 43 we do highlight one aspect of these critiques, namely the "risk of marginalised communities being denied access to structural measures". In the revised manuscript we will add other critiques. Also, we give a definition of building resilience, as "addressing and mitigating the complex interaction of social and economic vulnerability of communities and supporting their way of preparing for, coping with and recovering after disasters". We feel that with this there is no need for an additional definition of resilience.

8. We totally agree with this point. On lines 404-405, we suggest long-term empirical research on the effects of methods: "One of the methodological challenges in evaluating effects and drivers is that change can happen many months after an intervention. Longitudinal studies and being embedded in the communities would be needed (Donovan, 2010)." In the revised paper we will expand the statement on line 175, but keeping in mind that this is the literature review section and we want to keep our opinion and suggestions until Section 4.

9. Another good point: we will add some reflections on different types of impact and how to measure these.

14. We are not sure what exactly is referred to in this comment and how it links to creative practice re-locating agency. Please let us know what you mean.

15. We'll clarify the research gap on line 328.

---

## Author Comment (AC2) · 22 Apr 2020

We want to thank Louise Arnal for reviewing our manuscript. Thanks for the positive words about our paper! In the next version of the paper we will address her comments. Here some initial answers to the main comments raised in RC1.

1. We agree with the point raised that the aim should be to "investigate the added value of creative practices within more traditional current processes". We have indeed suggested this in our list of suggestions, but will also clarify this in the rest of the manuscript.

2. We will also elaborate on how creative practice can be used in conjunction with other (more traditional) methodologies. For example, we will discuss more on the aspect of longevity of effects (as also suggested by Zareen Bharucha in SC1) and highlight more the cultural embeddedness of the creative practice. With regard to the former we do want to point out that in this study we are not focussing on art as a product, but more on art as a process. This means that we are not looking at the artistic product themselves being timeless, but that we are more wondering how long-lasting the effects of engaging in artistic practice on resilience are. With regard to the latter we will point out that this embeddedness is important just like traditional engineering flood and drought measures also only work if they are tuned correctly to the local circumstances (both natural, socio-economic and cultural).

3. We had another look at these paragraphs and agree that some clarification would be beneficial. We included them in the introductory section of the literature review, because they include references outside those identified in the systematic literature review. Therefore, we would not prefer to move these paragraphs to the next sub-section. We will, as the reviewer suggests, clarify how these examples fit the classification introduced before.

4. We agree that there are some very good examples of using games in decision making on water-related issues. For this paper, we see these as outside the scope of our review. In the revised version of the manuscript we will add some text mentioning the examples and explaining that we explicitly excluded games.

5. We will implement the suggestions for improving Fig.4. The lighter colour for CS and F5 are meant to indicate that besides their main audience there is a secondary audience envisioned. This will be clarified in the caption.
   With regard to the graphic format of Fig.4 we tried different options, for example a 3D space, but this was even harder to read. We also considered a pie chart as suggested by the reviewer and although this looks nice, it would give 3x 8 = 24 graphs (figure below shows this for only 2 papers). Also it suggest an accuracy in the classification that we cannot claim. In the end we decided to keep the table format of Figure 4 with slight changes to the layout and a much clearer explanation in the caption and text.
   It would indeed be very interesting to also map the wider literature into these categories for comparison, but this would be an enormous task, because it requires carefully re-reading all 267 journal articles selected during the literature review and manually classifying them into the categories.

[Figure]

| goal | | doer | | | audience | | | | |
|---|---|---|---|---|---|---|---|---|---|
| awareness | action | community | academic | artist | participants | others in community | decision makers | general public | researchers |
| F1 | 60% | 40% | 20% | 80% | 0% | 50% | 50% | 0% | 0% | 0% |
| F2 | 100% | 0% | 0% | 50% | 50% | 10% | 0% | 0% | 80% | 10% |

6. Thanks for the suggestions for chapter 4.
   - We actually did search for creative practice examples outside the scientific literature and agree that a lot can be found there. We decided however not to include these in this paper as we found that the search methodology was less robust (much more dependent on Google search terms). But we will mention this in the revised manuscript.
   - We have tried to summarise the challenges we encountered in the case study (which often were also reflected in the literature) in lines 341-352 and mentioned some lessons-learned in lines 353-355. In the revised manuscript we will expand this a bit and make it more specific to our case study experience.
   - Since we only had funding for a short pilot project, we unfortunately cannot evaluate the longer term effect of our project. This is a common problem with most funders funding implementation of innovative ideas and not necessarily its longer-term impacts. We will make this more explicit in the revised version of the manuscript.

In response to the specific comments, we will define Global South communities, give examples of creative practice and artistic artefacts, and explain the critiques of the term resilience (see our response to SC1). The issue with the goals (P3 L63-64 and P4 L112) relates to point 3 above. The combination of goal, doer and audience leads to different types of using artistic practice in resilience research. Hopefully with rephrasing the examples in sub-section 2.1, this becomes clear.

The other specific and textual comments will be addressed in the revised version of the manuscript.

---

## Short Comment (SC2) · 1 May 2020

During a recent virtual writing retreat, we used a peer-review framework to review your abstract. We then had an open discussion and noted down all the feedback. We reviewed your abstract using a structured worksheet with the following advice in mind: "The abstract is a condensed and concentrated version of the full text of the research manuscript. It should be sufficiently representative of the paper if read as a stand-alone document". We looked for important elements of a research abstract and we comment on them below. We hope the following is helpful for your revisions. It's important to note that Geoscience Communication puts a lot of emphasis on evaluation of communication practice and ensuring that the practice is based on a solid research question and research design. The articles need to tell the story of research on geo-

science communication and not just tell the story of geoscience communication that's been done.

Overall: Your project sounds very interesting and we were really interested in the "creative process" and how it can be used with communities. The abstract touches on some very interesting elements and issues, which made us want to read on. However, there are a few things we believe should be improved for this to be relevant for a peer-reviewed publication.

Title: The title is nice, but we felt it referred more to the "literature review" part of the study than the pilot study. We felt that the pilot study should be emphasised more. The authors use the word "potential", but they have in fact carried out a pilot study where they put some of these ideas to the test. That sounds more like an "actual" tool rather than a "potential" tool.

Need and relevance: You have communicated the need and relevance in the first couple of sentences, but we got lost a little in the long second sentence. We noted repetition in this list that could be reduced. For example, the first and second points seem repetitive as well as the third and fourth. We also lacked some kind of definition of a "creative practice" to latch on to. This is described in slightly more detail further down, but maybe you could integrate a clear (but short) definition into the initial need and relevance part of the abstract.

Question/Hypothesis There seems to be 2 parts to this paper which comprise of a literature review and a pilot study. The questions/aims for these parts are split up. We suggest that the authors draw these together and connect them into a single clear question/aim/hypothesis for the whole paper. In a way, the literature review provides a clear question/need. Maybe consider leaving out the details of the review and simply say that a "literature review provides us with an important question:. . ."

Methods: The methods presented are a literature review and a pilot study. Firstly, we were uncertain from the abstract whether you have performed the literature review

yourselves or not. If you did, then you should use the active voice and "own" this part of the study more clearly. We also lacked a clearer description of exactly which "creative practice" the authors used in the pilot study. For us, this is a very important part of the story the abstract lacks. Maybe cut down on the literature review part and add a sentence explaining the "creative practice" implemented in the pilot study.

Results and conclusions There are hints of some interesting results, but they are not clearly communicated. Geoscience Communication puts a lot of emphasis of evaluation of communication practice. In this regard, writing "we noticed..." does not imply a thorough evaluation. We are left wondering how the authors actually measured the effect of the pilot study. You might consider deleting the sentence where you say the "effects not being thoroughly researched" and concentrate on communicating some concrete results. There must be some!

Take-home message The abstract ends with a couple of sentences that basically repeat the first sentences. We also do not recommend that the abstract contains a "we need more research" statement. The abstract could end more powerfully by simply deleting the last two sentences and saying something more concrete about what exactly the pilot study achieved.

Clarity The abstract reads quite nicely but there is some repetition that should be dealt with. For example, the last sentence basically repeats the "need". Also the description of the literature review uses 3 sentences and could be cut back to leave more room for a description of "creative practice", the results and a more powerful take-home message.

One of the peer-reviewers in our group constructed the following suggestion to help the authors understand some of the issues we had:

"Global South communities are increasingly exposed and vulnerable to natural hazards such as floods and droughts. Preparing for future extremes requires diverse knowledges, enhancing communication between diverse groups, and instigating organisational and behavioural change. Art and creativity (DESCRIBE THIS BET-

TER) are often used for raising awareness of climate change and for encouraging behavioural change in relation to health issues. Research using creative practice to increase resilience to natural hazards is still rare and it is important to investigate how the use of creative methods compares to other methods. SENTENCE WITH THEIR HYPOTHESIS. In this study we investigate if artistic and creative processes support Global South communities in improving their preparedness to extremes. We have done a literature review of 267 journal articles published between 2000 and 2018 which shows that there is a growing body of research on using creative practice in environmental issues. Furthermore, we carried out a pilot project where we tested the application of art and creativity in South Africa. We designed storytelling workshops to create community narratives about impacts of, and preparedness for, future drought. These narratives were filmed and edited and shared both with the community and governance actors. CONCLUSION SENTENCE ON THEIR WORK." We really enjoyed peer-reviewing this interesting abstract and commend the authors on what seems to be a very interesting study. We hope that our suggestions help to revise the abstract so that the essence of the research is communicated in a clearer and more powerful way. All the best, Mathew Stiller-Reeve (Thematic Editor of Geoscience Communication) and 6 peer-reviewers from The National Graduate School in Infection Biology and Antimicrobials in Norway.

Please also note the supplement to this comment:
https://www.geosci-commun-discuss.net/gc-2020-11/gc-2020-11-SC2-supplement.pdf

---

## Referee Comment (RC2) · Susanne Maciel (Referee) · 3 May 2020

This paper brings a very interesting review on creative practices in environmental issues, systematizing a large number of papers on art-based projects designed to foster awareness, behavioural change and communication between different groups. It also describes a pilot project in South Africa, where storytelling workshops were conducted to create community narratives about impacts of and preparedeness for future droughts. I really enjoyed reading this paper, and in my opinion the paper is suitable for publication in Geoscience Communication. I would like to point out some considerations about the narrative and some minor issues throughout the text, that the authors and editors can feel free to incorporate into the final publication:

[Figure]

1) General comments

- The authors mention their project in South Africa as a pilot project, and they fit it in the narrative as filling a gap identified in the reviewed literature, when using creative process to promote awareness on floods and droughts issues in a co-production approach. In my opinion, the project should be described more as a complete research project rather than a pilot one, and include more aspects on the materials and methods, such as including pictures of the models that were presented to the community, and describing with more details how the workshops were conducted. This might be useful information for other groups that are conducting similar research.

- Still about the project in South Africa, it was unclear to me what is the relation between the researchers and the community. Why and how that specific community was chosen, and how the researchers are related to the community are important aspects when dealing with marginalized communities.

- The authors recognize the sensitivity of the research topic, when dealing with marginalized communities that are often denied access to structural measures (L42-45). This makes me confort to point out some narrative aspects that might reinforce a hegemonic view on the subject. For example, it is repeated several times throughout the text that creative practices are useful to elevate hidden voices, but I think the authors should make it explicit to whom these voices are hidden. Also, the authors make an analysis of increasing number of papers in the literature focusing on art-based and creative practice in the reseacrh field of environmental and health issues. I was struck by the fact that "most art-based reseach is carried out in the United States, Canada and the United Kingdom" (L129-130)", and that Africa is the continent where most of these research projects are conducted. I am aware of several art-based projects happening in Latin America, by Latin American researchers. I can imagine that the same happen in other continents, and that these projects are usually not published in English. My point is that the authors could make a comment on the fact that since the keywords used by the authors are English words, you are automatically excluding a large number of papers written in other languages, and this turns your analysis biased. I don't think this is a problem, but I think it should be explicited in the text.

- I personally don't like the term "Global South", and I will explain why. According to Wikipedia, the term 'Global South' originated in postcolonial studies, and was first used in 1969. The term is highly used from 1980 till 2008, and even more afterwards, to define the set of countries that are poor, less-developed or oppressed and powerless. First of all, the term is inaccurate, because it refers to economic development notion by a geographic term. In this case, it includes communities in the North (L33), which is confusing and vague. Secondly, it homogenizes all countries in the southern hemisphere, and obscures important differences between them. The authors mention that "most researchers do not use the keyword 'Global South' in their titles or keywords" (L436), and I don't think this is a coincidence or lack of knowledge of the term. If the authors want to mention "poor communities", or "less developed communities", they should use these words, instead of highlighting a confusing term such as Global South. I highly recommend the authors to rethink the usage of the term.

- I agree with Louise Arnal's comments about Figure 4. I also found it difficult to follow. I would suggest that Table 1 be presented before Figure 4.

- The authors emphasize the need of evaluate the impacts of creative practice projects. In my experience, creative practice approaches usually show results in long term actions, and maybe this is an aspect that the autors could explore a little bit more in their literature review. The authors argue that the papers reviewed often describe a methodology without clear evaluation of its efficacy (L339), but how many of these papers describe long term projects?

- Still about evaluation, I would like to read how art-based projects usually evaluate their results, when they do. An interesting example is described by Strickert and Bradford: Of Research Pings and Ping–Pong Balls: the use of forum theatre for engaged water security research, International Journal of Qualitative Methods, 14, 1–14,

https://doi.org/10.1177/1609406915621409, 2015.) They use the Forum T heater to engage the community and policy makers for water security issues, and they evaluate the impacts of the approach by analysing the audience interferences into the play for each scenario. It is a very interesting example of evaluation of how creative practice can foster communication between researchers, community and policy makers.

- Just for the sake of knowledge, and perhaps to complement the analysis made in L39-40, I also would like to mention some works that are carried out in Brazil, where long-term theater based projects are conducted with rural populations, that might interest the authors: Boas, R. L. V., Pinto, V. C., and Rosa, S. M.: The School of Political Theater and Popular Video of Federal District: formation by praxis, Urdimento, 1, 36–47, https://doi.org/10.5965/1414573101342019036, 2019. and Gomide, C. S., Villas Boas, R. L., Martins, M. L., Gouveia, L. R., and Dias, A. L.: Rural Education and Pedagogy of Alternance: UnB experience in the Kalunga historical site and cultural heritage, The Brazilian Scientific Journal of Rural Education, 4, 1–27, https://doi.org/10.20873/uft.rbec.e7187, 2019.

- I L303, the authors mention that "in the workshops the narrative approach supported by data from the model scenarios allowed participants to use their imagination and exchange ideas". I think this is a really important result of your work, and should be more explored in the text.

2) Specific comments

- Could you please give some reference on the usage of the term Traditional Ecological Knowledge? (L70)

- Could you please specify how did you inferred the "preference for storytelling compared to other (more visual) methods"? (L230)

-L255: "attendance was low for some groups". How low?

- How participants to the workshop were selected?

- In L273: Could you explain why and how did you inferred that the community members did not show understanding of how different types of drought were linked and space and time?

- In L296 the authors mention it was more difficult to communicate about scenarios related to human activities. Could you please explain why?

- I would like to see references on the usage of the SHETRAN model, and if possible, some figures of how these models were presented to the community.

-L324: Do you think your images could lead to biased illustrations of the community to the policymakers? Why?

3) Technical corrections

-L4 – In this study, (add a comma)

- L7 Art and creativity are for example often used for raising awareness of climate change and for encouraging behavioural change in relation to health issues. (change in relation to → concerning?)

- L14 These kinds or this kind of methodologies.

- L354 - Remove spaces between practioners/artists

- L32, Fig 4 caption, L194, L195, L330, L396 and L414 – Add an hyphen in end-product

- L247 3oC

-L358 Change "can lead to fatalistic views that are not helpful for instigating" to "can lead to fatalistic views that do not help instigate"

- L436 – Change specify to specified

- L437- 438 – Add an hyphen in Middle-Income and High-Income -L437 – Remove preposition "and vulnerable (indigenous) group

---

## Author Comment (AC3) · 3 Jun 2020

We want to thank Mathew Stiller-Reeve and the 6 peer-reviewers from The National Graduate School in Infection Biology and Antimicrobials in Norway for commenting on our manuscript. We are very grateful for their suggestions.

In the next version of the paper we will address their comments. We will follow their suggestion to remove the word "potential" from the title and we will include the overall research aim (currently mentioned in the Introduction) in the abstract. We will also add the definition of 'creative practice' (see below) and work on the overall clarity of the abstract.

"Where creativity can be seen as the production of a novel and appropriate response

[Figure]

to a particular concern (Sternberg, 1999), creative practice is commonly associated with arts-led techniques. The term stretches to cover writing, theatre, dancing and a host of other imaginative activities (Light et al. 2018), not all of which need to result in a conventional product of aesthetic merit (Field, 1950)."

References

Field, J [pseudonym for Marion Milner] 1950 On not being able to paint Heinemann, London.

Light A, Mason D, Wakeford T, Wolstenholme R & Hielschler S (2018) Creative practice and transformations to sustainability: making and managing culture change. https://connected-communities.org/wp-content/uploads/2018/08/Creative-Practice-and-Transformations-to-Sustainability-Making-and-Managing-Culture-Change.pdf accessed 18/5/20

Sternberg R, ed. (1999) Handbook of creativity Cambridge University Press, Cambridge

---

## Author Comment (AC4) · 3 Jun 2020

We want to thank Susanne Maciel for reviewing our manuscript. Thanks for the positive words about our paper! In the next version of the paper we will address her comments. Here some initial answers to the main comments raised in RC2.

1) General comments:
- We are happy to provide more **details of the pilot study** and answer the reviewers questions. For example, we will describe the choice of community and how we relate to the community. We will also add a bit more information on the modelling, but we will be referring the reader to the paper published about this part of the research (Rangecroft et al., 2018) for more details. We do not think it is appropriate to add figures of the model results in this paper, since they would distract too much from the main message of this paper and they are already provided in Rangecroft et al. (2018). We will add some more detail on the workshops and update the reference to the part of the project working with policy makers (Makaya et al., 2020). However, we prefer not to present our pilot study as a complete research project in this paper, partly because this would upset the balance between the two parts of the paper (the literature review and the pilot study) and partly because the results of the research are discussed in other publications (Rangecroft et al., 2018; Makaya et al., 2020; Rohse et al., in prep), and so we cannot fully reproduce this information.
- Thanks for pointing out the **language bias** in our search. We fully agree that we are missing a lot of art-based research on this topic written in other languages. We have addressed the issue of language when discussing the pilot study, but not the literature search, which is an important omission. We will add this to Section 4 Reflections & Perspectives.
- We agree that the use of the term **'Global South'** to denote poor, less-developed or marginalised communities has geographic connotations that are confusing**.** However, the term 'less-developed' equally has important downsides. For example, using the terms developing / less-developed has connotations of a western standard of development and a narrow focus on economic growth. There appears to be no consensus within the scientific community which term is best, but we do find that the term Global South is widely used. Global South as a term has developed from a merely geographical to more of a political and economic characterisation. It is now generally used as relating to, but not completely overlapping with issues of inequality, power, poverty. It therefore encompasses a variety of groups in society, including rural populations, those in informal settlements, indigenous peoples, marginalised groups because of race, gender, age. These groups can be located in the geographic South or North, see Mahler (2018): "there are Souths in the geographic North and Norths in the geographic South" (Mahler, 2018: 32). Although we do recognise that the term has limitations, we suggest to stick with Global South in this paper. We will, however, add a sentence explaining our choice for the term in the revised version of the manuscript.
- **Figure 4**: We have looked again at a different way to visualise the results depicted in Figure 4. We are now suggesting to use a spider / polar diagram, which addresses the points made by both reviewers (Louise Arnal & Susanne Maciel). The draft figure below shows all papers in a polar diagram with three axes. The GOAL axis is gradually going from Awareness close to the centre to Action at the top and a combination in between. The DOER axis has Community close to the centre, two points for artist and researcher at the left-lower end and co-creation in between. The AUDIENCE axis has five points on the axis: Participants, Others in the community, Decision makers, General public and Researchers. All papers have different colours, the Drought papers in red and the Flood papers in blue and green. The Case Study is in orange and on the AUDIENCE axis it spans a few categories: from Participants, to Others in community, to Decision makers. The location of the papers on these axes is of course not accurate but a best estimate. We feel that in this way the results are much easier to grasp in one glance. You can now see the connection between the different axes of one paper, which was very difficult before. Also, the two categories we

explain in the paper (page 9-10, lines 196-199) now show more clearly, namely the papers focussing on action mostly have the participants themselves as audience (D2, F3) and those focussing on raising awareness with the audience being the general public (D1, F2) or researchers (F4, F5).

We will develop this manually drawn version of the figure to a neat figure programmed in R to be included in the manuscript. We will also move Table 1 to be presented before Figure 4.

[Figure]

- In the revised version of the manuscript (in Section 4 Reflections & Perspectives) we will indeed discuss how effects of creative practice will often will only become visible in the long term and add a few examples of **how effects are evaluated** if they are. Thanks for the paper suggestions.

- We agree that how **'the narrative approach supported … participants to use their imagination and exchange ideas'** is an important part of the results of our work. These aspects will be explored in more detail in another paper that is under development (Rohse et al., in prep). In the revised version of this paper we will add a few more observations on how the narrative approach supported imaginative exploration and exchange of ideas.

The specific and textual comments will be addressed in the revised version of the manuscript. We will provide a reference on the usage of the term 'Traditional Ecological Knowledge' and more details on the methodology used in the pilot study as requested by the reviewer.

**References**

Mahler, A.G. (2018) From the Tricontinental to the Global South: Race, Radicalism, and Transnational Solidarity. *Durham: Duke University Press*.

Makaya, E., Rohse, M., Day, R., Vogel, C., Mehta, L., McEwen, L., Rangecroft, S. & Van Loon, A. F. (2020). Water governance challenges in rural South Africa: exploring institutional coordination in drought management. *Water Policy*.

Rangecroft, S., Birkinshaw, S., Rohse, M., Day, R., McEwen, L., Makaya, E., & Van Loon, A. F. (2018). Hydrological modelling as a tool for interdisciplinary workshops on future drought. *Progress in Physical Geography: Earth and Environment*, *42*(2), 237-256.

---

## Author Response (AR2)

Dear editor,

Please find attached our revised manuscript (normal version and tracked-changed version) and below our point-to-point response to both reviewers and community comments (in bold, with original reviewer comment in normal font). We have made two new figures (Fig. 4 & 5), rephrased the abstract and large parts of the Introduction and Reflections sections, and added many details on the pilot study. We hope we have now adequately addressed all concerns and suggestions and are looking forward to hearing your (positive) response about our manuscript.

Best regards,

Anne Van Loon (on behalf of all authors)

**RC1**

**We want to thank Louise Arnal for reviewing our manuscript and for the positive words about our paper. In the revised version of the paper we addressed her comments. Below our point-to-point response to the questions raised in RC1, with page and line numbers referring to the tracked-changed version of the revised manuscript.**

*Main comments:*

1. The authors seem to suggest throughout the paper that the impacts of using creative practice should be analysed against using more traditional methodologies. E.g. on P1 L16-17 and P16 L398-399. In my opinion however, and as raised by the authors in the last bullet point on P16, creative practice should be used in combination with more traditional methodologies. As such, the aim should not be to compare the impact of both processes, but rather to investigate the added value of creative practices within more traditional current processes.

   **>> We agree with the point raised that creative practices and more conventional processes should be combined. In the earlier version of the manuscript we did indeed discuss this in Section 4 and suggested it in our list of suggestions, but have now also mentioned it in the abstract (l.29-30), at the start of the manuscript (l.82), added a few sentences in Section 2 (l.285-289) and Section 4 (l.459-460 & l.562-566), and rephrased the conclusion (l.630).**

2. As a succession to this point, it would be great if you could strengthen your point on how creative practice can complement more traditional methodologies, perhaps in the discussion. This is very nicely reflected in the methodology you follow for the pilot project, as explained on P10 L228-229: the use of model outputs to prompt the participants' imagination. A few questions it would be great to have your opinions on are: 1) How can creative practice tackle the point you raise on P1-2 L23-26? E.g. you mention on P15-16 L375-379 that creative methods can help increase resilience if previous extreme events happened a long time ago or for future events outside of previous experience (which is expected to happen more frequently with climate change). 2) Another point is that the success of increasing dialogues between groups in a community is better measured on the long-term (P10 L216-217). Could you please reflect on how creative practice has a role in insuring this "longevity" of the success of a project. E.g. I would argue that art is timeless, whereas other more traditional methodologies might not be. 3) As hinted by your comment on P11 L230, different art forms appeal to different communities given their culture. E.g.

some communities might prefer storytelling, others dancing, etc. as these are art forms already deeply rooted within their culture. This is where creative practice can help over more traditional methodologies, by echoing a community's culture

**>> In the revised manuscript we have now elaborated on how creative practice can be used in conjunction with other (more conventional) methodologies. For example, we discuss more on the aspect of longevity of effects (as also suggested by SC1) and highlight more the cultural embeddedness of the creative practice. With regard to the former we do want to point out that in this study we are not focussing on art as a product, but more on art as a process. This means that we are not looking at the artistic products themselves being timeless, but that we are more wondering how long-lasting the effects of engaging in artistic practice on resilience are. With regard to the latter we now point out that this embeddedness is important just like conventional engineering flood and drought measures also only work if they are tuned correctly to the local circumstances (both natural, socio-economic and cultural) (l.548-559).**

3.  I found it interesting to read about the different examples of creative practices and their goal, doer and audience on P3-4 L70-110, but found it hard to understand the exact purpose of these paragraphs. It appears to already be a part of the literature mapping and hence might fit better in the next sub-section? If their purpose is to give examples of different combinations of the three dimensions you put forward earlier, it would be helpful to clarify this. It might also help guide the reader to be consistent and use the same language as introduced earlier regarding the three dimensions within these five example paragraphs. E.g. For the first point, the goal is to raise awareness by passing on knowledge between generations, the doer and audience are the community.

**>> As the reviewer suggests, we now clarify how these examples fit the classification introduced before (l.108-109, 116-118, 119-120, 132, l.140-141, 148-150). We still include them in the introductory section, because they include references outside those identified in the systematic literature review.**

4.  Where do games stand in the midst of the creative processes you looked at? I would argue that they are a creative practice. However, there is very little mention of games until P11 L230 (if I'm not mistaken). On P4 L109-110 you say that there is no example to your knowledge of artistic products in decision-making. I think that there are plenty of resources on the use of games in decision-making which it would be great to highlight. E.g. the numerous workshops organised by the Red Cross using "serious games": https://www.climatecentre.org/resources-games/games, the IHE Delft games: https://www.un-ihe.org/serious-games-decision-making, or the HEPEX games: https://hepex.inrae.fr/resources/hepex-games/.

**>> We agree that there are some very good examples of using games in decision making on water-related issues. For this paper, we see these as outside the scope of our review. In the revised version of the manuscript we have added some text to the Introduction (l.49-54) and citing a number of excellent (review) papers.**

5.  I found it hard to understand Fig. 4 and visualise the results you mention on P9-10 L196-203. I think another graphic format may be more suitable to highlight these results and the gaps in the existing literature. Pie charts may be more intuitive? Could you please also change "Method" to "Doer", to be consistent with language introduced on P3 L63. It is not clear to me why 2 of the CS and one F5 circles are lighter in colour. Additionally, I think it would be

very interesting to be able to map the wider literature onto this graphic to see how the flood- & drought-related literature compares to it.

**>> We implemented the suggestions for improving Fig.4. With regard to the graphic format of Fig.4 we tried different options and decided use a spider diagram. It would indeed be very interesting to also map the wider literature into these categories for comparison, but this would be an enormous task, because it requires carefully re-reading all 267 journal articles selected during the literature review and manually classifying them into the categories.**

6. Could you please reflect in the "reflections & perspectives" on: 1) how your findings may be limited by the mapping methodology you used: searching for research papers. There are surely a lot more creative practice examples out there, less researchoriented and with different goals, audience and doer, but not mentioned in any research paper. 2) What worked very well and less well in your pilot project: would you do anything differently now? What tips would you give to people who want to create such projects? 3) How do you foresee the continuation of the project? Do you plan to put in place methods to evaluate the long-term impacts this project may have on the community and decision-makers?

**>> Thanks for the suggestions for chapter 4.**

- **We actually did search for creative practice examples outside the scientific literature and agree that a lot can be found there. We decided however not to include these in this paper as we found that the search methodology was less robust (much more dependent on Google search terms). We now mention this in the revised manuscript (l.467-470).**
- **In the previous version of the manuscript we already summarised the challenges we encountered in the case study (which often were also reflected in the literature) in lines 341-352 (previous version) and mentioned some lessons-learned in lines 353-355 (previous version). In the revised manuscript we added a specific subsection with limitations of our pilot study (l.476-484) and included more examples from our pilot study throughout the Reflections section.**
- **Since we only had funding for a short pilot project, we unfortunately cannot evaluate the longer term effect of our project. This is a common problem with most funders funding implementation of innovative ideas and not necessarily its longer-term impacts. We now make this more explicit in the revised version of the manuscript (l. 484) and discuss the aspect of timescales for evaluation (l.503-512).**

*Specific comments:*

In response to the specific comments, we will define Global South communities, give examples of creative practice and artistic artefacts, and explain the critiques of the term resilience (see our response to SC1). The issue with the goals (P3 L63-64 and P4 L112) relates to point 3 above. The combination of goal, doer and audience leads to different types of using artistic practice in resilience research. Hopefully with rephrasing the examples in sub-section 2.1, this becomes clear.

- P1 L20: Could you please define here what you mean with Global South communities, perhaps by moving the definition on P2 L33-34 here. **>> DONE, thanks**.
- P2 L25-28: These 2 sentences seem like a repetition. Please considering merging and/or reformulating. **>> We merged these sentences.**
- P2 L31-32: This statement puzzled me at first. Could you maybe give examples of creative practice (with and without end products) and of artistic artefacts. **>> We rewrote this sentence and added a few examples.**

- P2 L37: Could you please clarify what you mean by "suitable traditional structural or non-structural measures". **>> We added some examples.**
- P2 L41: Could you please explain briefly what the main critiques of the term "resilience" are. **>> We expanded on the most relevant critique for this paper, but for additional critiques we refer to the papers mentioned.**
- P3 L63-64: The examples of goals you mention here do not correspond to the goals you mention on P4 L112. Please consider updating this list, as well as throughout the paper. **>> We rephrased the latter sentence, also based on the changes made to the earlier paragraphs with the examples of the combinations of the three dimensions. We hope this is clearer now.**
- P3 L73-74: Please specify what is the purpose of the practice described by McEwen et al. **>> This has been added.**
- P4 L93: I would argue that here the audience would also be the end-users of the research product, for example the readers of the research paper if the research is published, who may or may not be researchers. **>> This is indeed the case for most categories. We now discuss this more clearly in the paper.**
- P4 L106: Could you please clarify what you mean by "mental models". **>> We have now added an explanation.**
- Fig. 1: If you have the data to plot this, it would be great to be able to visualise the separation per wider topics as well (i.e. hazards and disasters, climate change, other environmental issues, health, social and economic inequality, violence and conflict; or broader topics even), as different colours/patterns on the bars, to see how these change over time. E.g. Are the natural hazard articles more recent, even if they make a small overall portion of all papers? **>> We agree that this would be interesting, but we currently do not have the data ready to plot this.**
- P6 L 140-141: Could you please clarify what is "Photovoice (or Photo-Elicitation Methods or Camera-User-Study)" for less familiar readers like me. **>> We added an explanation of Photovoice.**
- P6 L 145-146: I understood this sentence only after having finished reading the whole paragraph. Could you please rephrase to clarify what is meant by "asking participants to develop new material". **>> We have added some text on the creation of new material in the previous subsection (l.117-118). This will hopefully make this sentence easier to understand.**
- P7 L166-167: Would you be able to give an estimate of the % for all environmental papers you looked at in the literature mapping? It would be nice to have it to compare the % with those for papers on droughts & floods later on, on P10 L200-201. **>> This figure was added (67%).**
- P7 L167: Could you please clarify what you mean by "medium or high" (also on P8 L169). **>> We added this information to the Appendix and added a reference to the Appendix in the text.**
- P8 L176: Please clarify that these papers are D1&2. **>> Done.**
- P8 L184-185: This is a repetition of the line on P8 L180-182. Please consider merging. **>> We rephrased this sentence.**
- P8 L186: Please clarify that these papers are F1-5. **>> Done.**
- P9 L192: Please specify here that this is referring to Fig. 4. **>> Done.**
- P9-10 L196-199: This is not clear to me. Are the 2 categories you mention within the "goal" and "audience" categories? If so, could you please rephrase these sentences. The choice of the words "Firstly" and "Secondly" may be confusing me. Could you please also discuss what

results are with regards to the "Doer" (or "Method"). **>> Yes, these two categories relate to the "goal" and "audience" dimensions. We clarified this (l.268). The doer dimension is discussed later, which we have now also indicated more clearly.**

- P10 L199: Is "instigating action" pre-disaster similar to "raising awareness"? **>> No, it is not. In these cases, the community would implement measures to be better prepared for flood/drought. We added an explanation.**
- P10 L201: It would be great if you could mention again here what the percentages are of studies with a co-creation aspect, for comparison, for: floods & droughts, health issues and environmental issues. **>> Done.**
- P10 L210: Could you please clarify what you mean by "experimentation". **>> This is taken from Biggs (2012) paper. We refer the reviewer and readers to this paper for further clarification.**
- P10 L212: You use the plural form of "approach" on P10 L207. Is this because the project was made of several approaches which your creative approach was a part of? Please clarify in the text. **>> We changed this to singular "approach".**
- P11 L230: This is very interesting and merits further discussion. It appears that participants preferred storytelling as it is an art form already rooted in their culture perhaps? It would be great if you could add a few lines in the discussion about how processes/media might not be transferrable across communities as different communities have different histories/sensitivities to different arts. This perhaps complements your second point on P16: the goal, doer and audience are situation-specific, but so is the creative form/media used. **>> We added a paragraph on this in the Discussion section (l.550-557).**
- P11 L240: Do you have any reference you could cite here for the SHETRAN model? **>> Two references added.**
- P11 L252-253: I imagine the phrasing of these scenarios was worked on with care as they could lead to different responses from the participants. Could you please say a bit more on how the storylines were written? And maybe give a written example of a model output and the attached storyline in the paper? **>> We added some more information on the storylines and an additional figure (Fig. 5). For more details we refer to the paper by Rangecroft et al. (2018).**
- P11 L254: Are the workshop categories in Phase 3 the same as in Phase 1? **>> Not completely, but using the same groupings. We explain this now.**
- P12 L287-289: Do you think this is just because of the age difference of these groups? These sentences would fit better in the former paragraph I think. **>> We expanded on this.**
- P13 L296-297: Please specify that you are referring for the workshops of phase 3. **>> Done.**
- P13 L311: Do you mean "request for government support"? **>> Yes, changed.**
- Fig. 5: It would be great, if possible, to have tangible narrative content in the paper as well. Only if possible, please consider adding perhaps parts of a narrative transcript, or a few quotes from several narratives, to the paper. **>> This is an interesting idea, but we feel that by adding one transcript we put too much emphasis on this one group's view whereas very different views were expressed during the workshops. The narrative transcripts will be explored in more detail in Rohse et al. (in prep.).**
- P14 L329-330: Could you please share opinions on why you think that is? **>> We added some thoughts on this.**
- P14 L333-335: I am not sure I understand this sentence. Could you please clarify. **>> We rewrote this paragraph.**
- P14 L336-340: I think you are being too harsh with yourselves. While you couldn't, as part of this project, evaluate the long-term benefits of your creative practice, there are some clear

short-term benefits that are worth mentioning again in the discussion. E.g. the fact that policymakers found the images you captured useful, the intergenerational exchanges your workshops led to, the fact that your workshops encouraged participants to use their imagination and exchange ideas vs referring to God in an uncertain future: : : These are already brilliant outputs which should feature here. **>> Thanks. We added this to this section.**

- P15 L371-373: This feels like an important point. Could you maybe answer this question using your pilot study? As you have seen both aspects in phase 1 and 3 of your workshops. **>> We added some evidence for this from our pilot study.**
- P15-16 L375-379: This is where creativity can foster exchanges across communities. E.g. A community who has not yet experienced a drought and may be confronted by these events in the future (for example due to climate change) could get an idea of what it is like on the ground by learning from communities facing droughts on a regular basis. **>> Thanks. We expanded on this point.**
- P16 L382-386: Could you please explain the purpose of this point further. **>> We added some text here.**
- P16 L404: Could you please clarify what "longitudinal studies" are. **>> We added the explanation.**
- P16 L396-405: Could the effectiveness of creative practice be measured by comparing different communities and how they have coped with environmental issues? For example aboriginal communities where the use of art seems to be deeply rooted in culture vs a culture where art is very rarely used. **>> Thanks. We added this point.**
- P17: Please summarise briefly what your paper is about before mentioning results. **>> We added a sentence.**

*Technical corrections:*

- P1 L21: Remove "and". **>> Done.**
- P1 L22: Replace "and" after "water demands" by a comma. **>> Done.**
- P3 L73: Add a dot after "environmental stress". **>> This is an enumeration, so the comma is needed.**
- P3 L73: "describe" without an s. **>> Done.**
- P4 L89: "built on" with a t. **>> Done.**
- P4 L111: Add "we" before "will". **>> Done.**
- P4 L112: "instigating". **>> Done.**
- P7 L156: "developed". **>> Done.**
- P10 L209: "redundancy" might not be the adequate wording. Do you mean "repetition"? **>> No, we mean redundancy. Biggs et al. (2012) write: "Redundancy is essentially the opposite of disparity and provides "insurance" for ES provision by allowing some system elements to compensate for the loss or failure of others."**
- P10 L217: "members' travel to". **>> We mean travel as verb here.**
- P11 L247: "3C". **>> Done.**
- P11 L247: Remove "a" in front of "climate". **>> Done.**
- P12 L294-295: "We used one climate change scenario and two scenarios related to human activities". **>> Done.**

**RC2**

**We want to thank Susanne Maciel for reviewing our manuscript and for the positive words about our paper. In the revised version of the paper we addressed her comments. Below our point-to-point response to the questions raised in RC2, with page and line numbers referring to the tracked-changed version of the revised manuscript.**

*1) General comments:*

- The authors mention their project in South Africa as a pilot project, and they fit it in the narrative as filling a gap identified in the reviewed literature, when using creative process to promote awareness on floods and droughts issues in a co-production approach. In my opinion, the project should be described more as a complete research project rather than a pilot one, and include more aspects on the materials and methods, such as including pictures of the models that were presented to the community, and describing with more details how the workshops were conducted. This might be useful information for other groups that are conducting similar research. **>> We are happy to provide more details of the pilot study and answer the reviewers questions. For more information on the modelling, but we refer the reader to the paper published about this part of the research (Rangecroft et al., 2018). We do not think it is appropriate to add figures of the model results in this paper, since they would distract too much from the main message of this paper and they are already provided in Rangecroft et al. (2018). We did, however, add a figure on the process of translating model results into storylines that were discussed with the workshop participants (new Figure 5). We also added some more detail on the storylines and the workshops (l.345-347 & l.396-397) and updated the reference to the part of the project working with policy makers (Makaya et al., 2020). However, we prefer not to present our pilot study as a complete research project in this paper, partly because this would upset the balance between the two parts of the paper (the literature review and the pilot study) and partly because the results of the research are discussed in other publications (Rangecroft et al., 2018; Makaya et al., 2020; Rohse et al., in prep), and so we cannot fully reproduce this information.**
- Still about the project in South Africa, it was unclear to me what is the relation between the researchers and the community. Why and how that specific community was chosen, and how the researchers are related to the community are important aspects when dealing with marginalized communities. **>> We have now described in more detail the choice of community and how we relate to the community (l.311-315).**
- The authors recognize the sensitivity of the research topic, when dealing with marginalized communities that are often denied access to structural measures (L42-45). This makes me confort to point out some narrative aspects that might reinforce a hegemonic view on the subject. For example, it is repeated several times throughout the text that creative practices are useful to elevate hidden voices, but I think the authors should make it explicit to whom these voices are hidden. Also, the authors make an analysis of increasing number of papers in the literature focusing on art-based and creative practice in the reseacrh field of environmental and health issues. I was struck by the fact that "most art-based reseach is carried out in the United States, Canada and the United Kingdom" (L129-130)", and that Africa is the continent where most of these research projects are conducted. I am aware of several art-based projects happening in Latin America, by Latin American researchers. I can imagine that the same happen in other continents, and that these projects are usually not published in English. My point is that the authors could make a comment on the fact that since the keywords used by the authors are English words, you are automatically excluding a large number of papers written in other languages, and this turns your analysis biased. I don't think this is a problem, but I think it should be explicited in the text. **>>**
    - **With regards to the surfacing of hidden voice: we do point out to whom they are hidden. For example, on l.72-73 we state that "According to Gibson et al. (2018),**

cultural resourcefulness and coping capacities of rural populations are rarely acknowledged within state-expert modelling of resilience." And we talk about how our videos were used more widely within the community and in conversations with policy makers. We have now strengthened this last point (l.302-304, l.434-435, l.480-483, l.526-527) and have also revised the wording throughout the paper.

- o Thanks for pointing out the language bias in our search. We fully agree that we are missing a lot of art-based research on this topic written in other languages. We have addressed the issue of language when discussing the pilot study, but not the literature search, which is an important omission. We added this now to Section 4 Reflections & Perspectives (l.471-474).

- I personally don't like the term "Global South", and I will explain why. According to Wikipedia, the term 'Global South' originated in postcolonial studies, and was first used in 1969. The term is highly used from 1980 till 2008, and even more afterwards, to define the set of countries that are poor, less-developed or oppressed and powerless. First of all, the term is inaccurate, because it refers to economic development notion by a geographic term. In this case, it includes communities in the North (L33), which is confusing and vague. Secondly, it homogenizes all countries in the southern hemisphere, and obscures important differences between them. The authors mention that "most researchers do not use the keyword 'Global South' in their titles or keywords" (L436), and I don't think this is a coincidence or lack of knowledge of the term. If the authors want to mention "poor communities", or "less developed communities", they should use these words, instead of highlighting a confusing term such as Global South. I highly recommend the authors to rethink the usage of the term. **>> We agree that the use of the term 'Global South' to denote poor, less-developed or marginalised communities has geographic connotations that are confusing. However, the term 'less-developed' equally has important downsides. For example, using the terms developing / less-developed has connotations of a western standard of development and a narrow focus on economic growth. There appears to be no consensus within the scientific community which term is best, but we do find that the term Global South is widely used. Although we do recognise that the term has limitations, we suggest to stick with Global South in this paper. We have, however, expanded the explanation of our choice for the term in the revised version of the manuscript (l.55-64).**

- I agree with Louise Arnal's comments about Figure 4. I also found it difficult to follow. I would suggest that Table 1 be presented before Figure 4. **>> We have changed Figure 4 into a spider / polar diagram, which addresses the points made by both reviewers (Louise Arnal & Susanne Maciel). We feel that in this way the results are much easier to grasp in one glance. You can now see the connection between the different axes of one paper, which was very difficult before. Also, the two categories we explain in the paper (l.268-271) now show more clearly, namely the papers focussing on action mostly have the participants themselves as audience (D2, F3) and those focussing on raising awareness with the audience being the general public (D1, F2) or researchers (F4, F5). We also moved Table 1 to be presented before Figure 4.**

- The authors emphasize the need of evaluate the impacts of creative practice projects. In my experience, creative practice approaches usually show results in long term actions, and maybe this is an aspect that the autors could explore a little bit more in their literature review. The authors argue that the papers reviewed often describe a methodology without clear evaluation of its efficacy (L339), but how many of these papers describe long term projects? Still about evaluation, I would like to read how art-based projects usually evaluate their results, when they do. An interesting example is described by Strickert and Bradford: Of Research Pings and Ping–Pong Balls: the use of forum theatre for engaged water security research, International Journal of Qualitative Methods, 14, 1–14, https://doi.org/10.1177/1609406915621409, 2015.) They use the Forum T heater to engage

the community and policy makers for water security issues, and they evaluate the impacts of the approach by analysing the audience interferences into the play for each scenario. It is a very interesting example of evaluation of how creative practice can foster communication between researchers, community and policy makers. **>> In the revised version of the manuscript (in Section 4 Reflections & Perspectives, l.486-531) we now discuss in more detail how effects of creative practice will often will only become visible in the long term and added a few examples of how effects are evaluated if they are (l.497-498).**

- Just for the sake of knowledge, and perhaps to complement the analysis made in L39-40, I also would like to mention some works that are carried out in Brazil, where long-term theater based projects are conducted with rural populations, that might interest the authors: Boas, R. L. V., Pinto, V. C., and Rosa, S. M.: The School of Political Theater and Popular Video of Federal District: formation by praxis, Urdimento, 1, 36–47, https://doi.org/10.5965/1414573101342019036, 2019. and Gomide, C. S., Villas Boas, R. L., Martins, M. L., Gouveia, L. R., and Dias, A. L.: Rural Education and Pedagogy of Alternance: UnB experience in the Kalunga historical site and cultural heritage, The Brazilian Scientific Journal of Rural Education, 4, 1–27, https://doi.org/10.20873/uft.rbec.e7187, 2019. **>> Thanks for the paper suggestions.**

- L303, the authors mention that "in the workshops the narrative approach supported by data from the model scenarios allowed participants to use their imagination and exchange ideas". I think this is a really important result of your work, and should be more explored in the text. **>> We agree that how 'the narrative approach supported … participants to use their imagination and exchange ideas' is an important part of the results of our work. These aspects are explored in more detail in another paper that is under development (Rohse et al., in prep). In the revised version of this paper we added a few more observations on how the narrative approach supported imaginative exploration and exchange of ideas (l.421-424, l.523-525).**

*2) Specific comments:*
- Could you please give some reference on the usage of the term Traditional Ecological Knowledge? (L70) **>> We included a reference.**
- Could you please specify how did you inferred the "preference for storytelling compared to other (more visual) methods"? (L230) **>> This is further explored in the Results section, where we stated that "We talked about visual methods like artistic maps or other methods like board games for the community to interact with potential future changes in water availability and use. However, from the start it was clear that the idea of 'stories' was most resonant with the community. The participants of phase 1 and the village elders of the royal council indicated their interest in developing stories." (l.380-383).**
- L255: "attendance was low for some groups". How low? **>> We rephrased this sentences and added the numbers.**
- How participants to the workshop were selected? **>> Participants were selected by the village leadership. This has now been added.**
- In L273: Could you explain why and how did you inferred that the community members did not show understanding of how different types of drought were linked and space and time? **>> We added an example.**
- In L296 the authors mention it was more difficult to communicate about scenarios related to human activities. Could you please explain why? **>> We added a possible explanation and refer back to the example mentioned earlier.**
- I would like to see references on the usage of the SHETRAN model, and if possible, some figures of how these models were presented to the community. **>> We added the SHETRAN references and added more details on the storylines of model results that were presented**

**to the community. We also added an additional figure. For more details we refer to Rangecroft et al. (2018).**

- L324: Do you think your images could lead to biased illustrations of the community to the policymakers? Why? **>> We added a sentence.**

*3) Technical corrections*
- L4 – In this study, (add a comma) **>> DONE**
- L7 Art and creativity are for example often used for raising awareness of climate change and for encouraging behavioural change in relation to health issues. (change in relation to ! concerning?) **>> DONE**
- L14 These kinds or this kind of methodologies. **>> CHANGED**
- L354 - Remove spaces between practioners/artists **>> DONE**
- L32, Fig 4 caption, L194, L195, L330, L396 and L414 – Add an hyphen in end-product **>> DONE**
- L247 3oC **>> DONE**
- L358 Change "can lead to fatalistic views that are not helpful for instigating" to "can lead to fatalistic views that do not help instigate" **>> CHANGED**
- L436 – Change specify to specified **>> DONE**
- L437- 438 – Add an hyphen in Middle-Income and High-Income -L437 – Remove preposition "and vulnerable (indigenous) group **>> DONE**

**SC1**

**We want to thank Zareen Bharucha for commenting on our manuscript. In the new version of the paper we addressed her comments and fixed the textual errors. Below our point-to-point response to the questions raised in SC1, with page and line numbers referring to the revised manuscript.**

1. **References are done automatically and cannot be changed at this stage. We will follow the journal guidance on this in the proofing stage.**
2. **As also suggested by SC2, we now provide a more complete definition of creative practice, including a few examples (l.44-53).**
3. **We mean marginalised communities, not (necessarily) rural communities. This encompasses vulnerable and socio-economically disadvantaged groups in society, which are more abundant in the South, amongst ethnic minorities in both South and North, and amongst more rural populations, but not exclusively. There might be marginalised communities in big cities and some rural populations might be wealthy and well-protected against natural hazards. We will clarify this distinction in the manuscript. Also based on the comment by RC2, we have adjusted and expended our description of Global South communities (l.55-64).**
4. **On line 43 we do highlight one aspect of these critiques, namely the "risk of marginalised communities being denied access to structural measures". In the revised manuscript, we added a sentence on the term resilience being used to mean 'self-reliance' by those in power (l.76-77). Also, we give a definition of building resilience, as "addressing and mitigating the complex interaction of social and economic vulnerability of communities and supporting their way of preparing for, coping with and recovering after disasters". We feel that with this there is no need for an additional definition of resilience.**
5. **Changed.**

6. **Changed.**
7. **Changed.**
8. **We totally agree with this point. In the previous version of the manuscript, we suggest long-term empirical research on the effects of methods: "One of the methodological challenges in evaluating effects and drivers is that change can happen many months after an intervention. Longitudinal studies and being embedded in the communities would be needed (Donovan, 2010)." In the revised paper, we added a cross-reference to Section 4 (l.245) and extended the paragraph discussing this aspect (l.493-512), also following suggestions by RC2.**
9. **Another good point: we added some reflections on different types of impact (l.504-507).**
10. **Thank you!** 😊
11. **Solved.**
12. **Corrected.**
13. **Changed.**
14. **We were not sure what the reviewer meant with this comment. But we have made some changes in this paragraph also based on the comments by RC2 (l.421-424, l.434-436, l.499-502). We hope this answers her questions.**
15. **We (re)clarified the research gap in Section 4 (l.438-445).**

**SC2**

**We want to thank Mathew Stiller-Reeve and the 6 peer-reviewers from The National Graduate School in Infection Biology and Antimicrobials in Norway for commenting on our manuscript. We are very grateful for their suggestions. In the new version of the paper we have completely rewritten the abstract addressing their comments. We followed their suggestion to remove the word "potential" from the title and added more results to the abstract. We also added the definition of 'creative practice' (see below) to the Introduction (l.44-54).**

[revised manuscript text omitted]